# Nanostructure-specific X-ray tomography reveals myelin levels, integrity and axon orientations in mouse and human nervous tissue

Marios Georgiadis [1,2,3✉], Aileen Schroeter[1], Zirui Gao [4,1], Manuel Guizar-Sicairos [4], Marianne Liebi [5], Christoph Leuze[3], Jennifer A. McNab [3], Aleezah Balolia [6], Jelle Veraart[2], Benjamin Ades-Aron[2], Sunglyoung Kim[2], Timothy Shepherd [2], Choong H. Lee [2], Piotr Walczak[7,8], Shirish Chodankar[9], Phillip DiGiacomo[3], Gergely David[10], Mark Augath[1], Valerio Zerbi [1], Stefan Sommer [1], Ivan Rajkovic[11], Thomas Weiss[11], Oliver Bunk [4], Lin Yang [9], Jiangyang Zhang[2], Dmitry S. Novikov [2], Michael Zeineh [3], Els Fieremans [2,13] & Markus Rudin[1,12,13]

Myelin insulates neuronal axons and enables fast signal transmission, constituting a key component of brain development, aging and disease. Yet, myelin-specific imaging of macroscopic samples remains a challenge. Here, we exploit myelin's nanostructural periodicity, and use small-angle X-ray scattering tensor tomography (SAXS-TT) to simultaneously quantify myelin levels, nanostructural integrity and axon orientations in nervous tissue. Proof-of-principle is demonstrated in whole mouse brain, mouse spinal cord and human white and gray matter samples. Outcomes are validated by 2D/3D histology and compared to MRI measurements sensitive to myelin and axon orientations. Specificity to nanostructure is exemplified by concomitantly imaging different myelin types with distinct periodicities. Finally, we illustrate the method's sensitivity towards myelin-related diseases by quantifying myelin alterations in dysmyelinated mouse brain. This non-destructive, stain-free molecular imaging approach enables quantitative studies of myelination within and across samples during development, aging, disease and treatment, and is applicable to other ordered biomolecules or nanostructures.

[1] Institute for Biomedical Engineering, ETH Zurich, Zurich, Switzerland. [2] Center for Biomedical Imaging, New York University School of Medicine, New York, NY, USA. [3] Department of Radiology, Stanford School of Medicine, Stanford, CA, USA. [4] Swiss Light Source, Paul Scherrer Institute, Villigen, Switzerland. [5] Department of Physics, Chalmers University of Technology, Gothenburg, Sweden. [6] Department of Integrative Biology, University of Colorado Denver, Denver, CO, USA. [7] Department of Radiology, Johns Hopkins Medicine, Baltimore, MD, USA. [8] Department of Diagnostic Radiology & Nuclear Medicine, University of Maryland, Baltimore, MD, USA. [9] National Synchrotron Light Source II, Brookhaven National Laboratory, Upton, NY 11973, USA. [10] Balgrist University Hospital, University of Zurich, Zurich, Switzerland. [11] Stanford Synchrotron Radiation Lightsource, SLAC National Accelerator Laboratory, Menlo Park, CA 94025, USA. [12] Institute of Pharmacology and Toxicology, University of Zurich, Zurich, Switzerland. [13] These authors contributed equally: Els Fieremans, Markus Rudin. ✉email: mariosg@stanford.edu

Myelination is an evolutionary milestone of the vertebrate nervous system, effectively "insulating" neuronal axons and enabling fast signal transmission, among other functions[1]. Myelin levels, nanostructural integrity, and myelinated axon orientations are important determinants of brain development and aging[2,3], and are affected in the majority of neurological diseases[4–6]. However, current clinical and experimental myelin imaging methods suffer from limitations. Myelin-sensitive magnetic resonance imaging (MRI) methods[7,8] lack specificity, since other nervous system structures, (macro)molecules, or chemical elements may affect the signal in a way similar to myelin[7,8]. On the other hand, high-resolution optical microscopy in combination with myelin-specific stains, the reference method for ex vivo tissue analysis, suffers from low tissue penetration due to light scattering, and therefore requires either restriction to thin sections or tissue clearing procedures, both of which introduce artifacts limiting the methods' use for quantitative whole-sample analyses.

Small-angle X-ray scattering (SAXS) combines the high tissue penetration of X-rays with specificity to periodic tissue nanostructures, based on the constructive interference of scattered photons according to Bragg's law of diffraction. Myelin's $d = 15–20$ nm structural periodicity[9–11] produces myelin-specific SAXS maxima, also validated by combined SAXS/electron microscopy investigations[9,12]. Agrawal et al.[13] used myelin-specific SAXS intensities from single projections to show increased myelination during rodent nervous system development, assuming rotational invariance, i.e. that the SAXS signal intensity is independent of sample orientation in space. Jensen et al.[11] applied filtered back-projection to the azimuthally averaged SAXS signal and determined myelin levels for two rat brain tomographic slices, also assuming rotational invariance. However, myelin layer periodicity occurs radially to the axon orientation, so corresponding SAXS signal intensity patterns can be strongly anisotropic[14], which needs to be considered for accurate myelin level assessment[11]. The recently developed SAXS tensor tomography (SAXS-TT)[15–18] incorporates signal anisotropy in the reconstruction, providing 3D reciprocal space maps for each voxel, thus enabling quantitative investigations of both isotropic and anisotropic nanostructures.

We perform SAXS-TT experiments on *intact* macroscopic nervous tissue specimens such as mouse brain, mouse spinal cord, human white matter, and cortex, and use tensor reconstruction on the myelin-specific signal to quantify myelin levels and determine myelinated axon orientation and fiber tracts non-destructively. We validate the method against myelin histology, and further compare it with three-dimensional (3D) histology and MRI measurements of myelin and axon orientations, showcasing SAXS-TT as a reference method for myelin and axon orientation imaging. We demonstrate the nanostructural specificity of SAXS-TT by separately reconstructing central and peripheral nervous system (CNS/PNS) myelin signals based on their distinct nanostructure periodicities ($d_{CNS} \sim 15–17$ nm, $d_{PNS} \sim 19–20$ nm), and quantify alterations in CNS myelin levels and nanostructural integrity in a murine model of dysmyelination, the shiverer mouse. Altogether, we propose a method for nanostructure-specific imaging of macroscopic samples, which can serve as a reference method for quantifying myelin levels, nanostructure integrity, and axon orientations within and across samples.

## Results

### Tomographic mapping of myelinated axons.
As a proof-of-principle experiment, we applied SAXS-TT to the perfusion-fixed brain of a C57BL/6 mouse. Raster-scanning the brain (Fig. 1a) produced scattering patterns on the detector that exhibited intensity peaks corresponding to the $d_{CNS} = 15–17$ nm CNS myelin periodicity (Fig. 1a–c and Supplementary Movie 1 for all scattering patterns along one scan line). These peaks reveal the presence of myelinated axons along the beampath as well as their orientations[14]. In line with previous animal and human nervous tissue experiments[10,19,20], the first myelin-related peak at reciprocal space coordinate $q = 2\pi/d_{CNS} \approx 0.37–0.42$ nm$^{-1}$ is barely detectable due to the form factor of the repeated structure[10], while the second-order peak at $q = 2 \times 2\pi/d_{CNS} \approx 0.74–0.84$ nm$^{-1}$ displayed the highest amplitude. We refer to this second-order peak as the "myelin peak". Analysis of scattering intensity at these $q$-values for every scanned point provided two-dimensional (2D) maps for each projection (Fig. 1d and Supplementary Movie 2 for all projections). The 2D map of the azimuthally averaged scattering intensity at the position of the myelin peak, Fig. 1d(i), shows the brain clearly outlined with myelinated areas enhanced, whereas the X-ray transmission map, Fig. 1d(ii), displays almost no contrast within the sample or with the water-based embedding medium. Analysis of the azimuthal (i.e. angular) anisotropy of the signal in each scattering pattern provided in-plane orientations of brain structures, Fig. 1d(iii–iv).

Tensor-tomographic reconstruction using the iterative reconstruction tensor tomography (IRTT) algorithm[18] resulted in a rank-2 tensor representation of the anisotropic 3D reciprocal space map for each brain voxel. Figure 2a shows the tensor trace, which corresponds to the myelin-weighted SAXS signal intensity in each voxel. The myelin-specific part of the signal was isolated for every segment in each scattering pattern by a background fit and peak-finding procedure (cf. "Methods" section). Tensor-tomographic reconstruction[18] of the myelin-specific part and subsequent use of Funk Radon Transform[14] resulted in tensors representing the myelinated fiber orientation distribution function (fODF) per voxel. This enabled generating 2D and 3D maps of the myelin content given by the tensor trace (Fig. 2b, c), of the myelinated axon fODF (Fig. 2d) and the main fiber direction (Fig. 2e, f). The latter in conjunction with tractography algorithms enabled generating fiber tracts corresponding to myelinated axons only (Fig. 2g). Myelin quantification throughout the brain also enabled detailed comparisons between regions of the two hemispheres (Fig. 2h).

### Myelin levels and tracts on mouse spinal cord.
SAXS-TT was also performed on a mouse cervical spinal cord (Fig. 3). Mouse spinal cord fiber orientations reveal the anisotropic fiber orientation distribution, which appears more pronounced in the white matter tracts located in the periphery of the cord. SAXS-TT-derived fODFs, represented by a rank-2 tensor for each voxel, were used to generate synthetic diffusion MRI signals as input for probabilistic tractography. Resulting tractograms revealed dense white matter tracts running in the cranio-caudal direction (blue), whereas gray matter tracts are predominantly oriented radially (red: left–right; green: dorsal–ventral). Moreover, incoming fibers from the posterior horns are seen entering the spinal cord (white arrows), with most of them either directly connecting to the posterior column or decussating, as expected for the dorsal column and spinothalamic tracts, respectively. In addition, the fibers associated to motor neurons projecting toward the upper limbs are seen at the positions of the anterior horn (orange arrows). It should be noted that the same fiber tractography applied to dMRI data did not capture these connectivity patterns in the anterior and posterior horns (Supplementary Fig. 2), highlighting SAXS-TT's ability to resolve directional signals from the few myelinated axons in gray matter that are responsible for long-range connectivity.

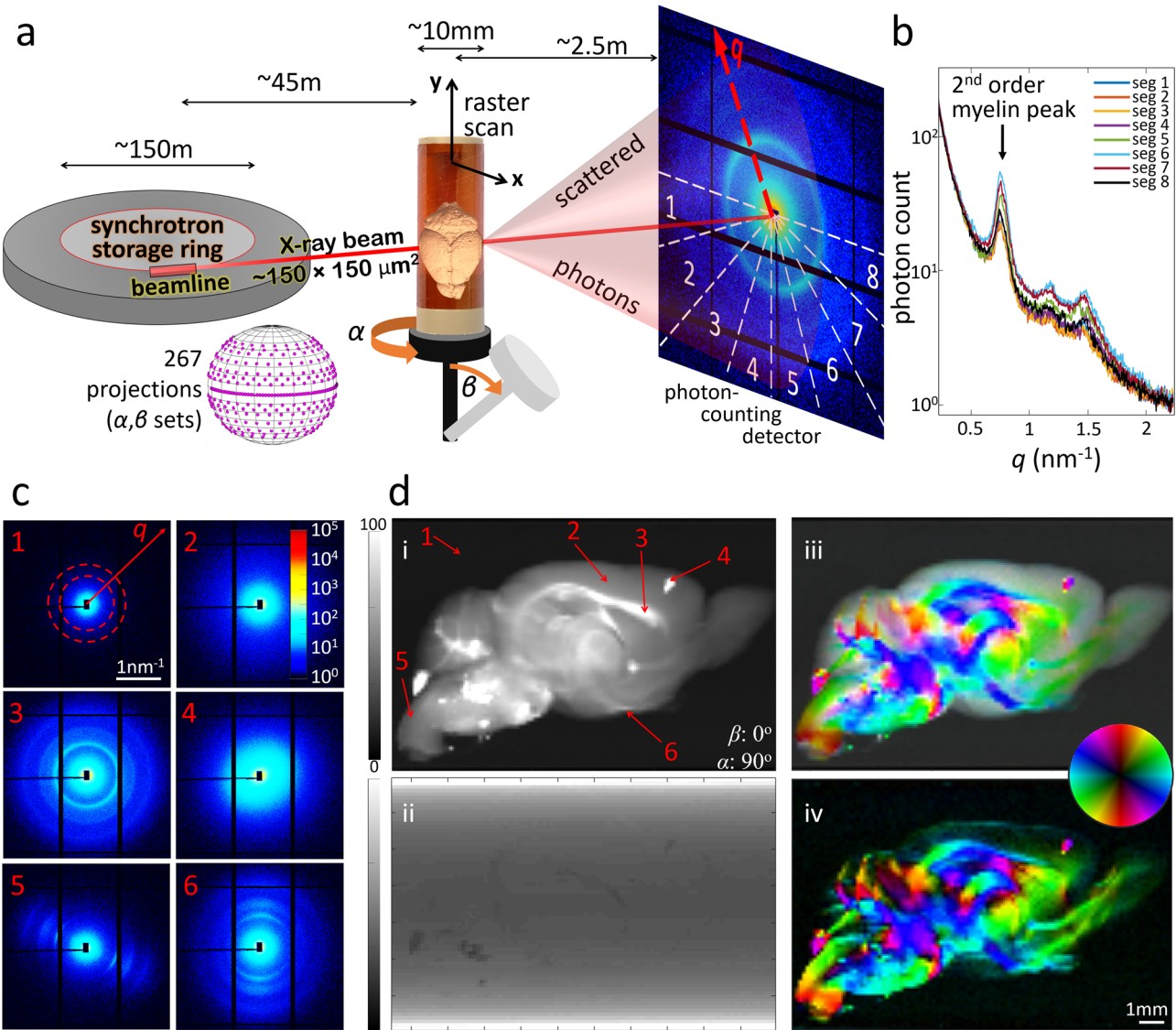

**Fig. 1 SAXS-TT experiments on mouse brain. a** SAXS-TT setup, with typical setup values. Real setup images from 3 synchrotrons in Supplementary Fig. 1. **b** Scattering intensity from one scan point against reciprocal vector q, for detector azimuthal segments in **a**. **c** Scattering patterns from points in **d** (i). Color bar: photon-count. Patterns: 1—embedding medium; 2—gray matter; 3, 5, 6—white matter; 4—bone residue on brain surface. **d** 2D maps for the $(\alpha, \beta) =$ (90º, 0º) projection, based on the signal along the beampath for each point. (i): azimuthally averaged myelin-peak scattering intensity, (ii): X-ray transmission, (iii, iv): color-encoded 2D orientation with (iii) and without (iv) averaged scattering intensity in (i). Color bars (i): photon-count, (ii): arbitrary units. In (iii, iv), hue-saturation represent 2D orientation-degree of orientation respectively.

**SAXS-TT on human white matter and cortex**. SAXS-TT on two human white matter specimens, splenium and body corpus callosum, from a 2-year-old female donor is presented in Fig. 4. Human white matter gave strong SAXS myelin signal (white arrows in scattering patterns) due to the high density of myelinated fibers. Most projections displayed little heterogeneity in axon orientations of both the splenium and the body specimens (also see Supplementary Movie 3 for all projections of the splenium scan) in agreement with the samples being mostly homogeneous in fiber orientations. The reconstructed axon orientations were used as input for tractography, with the resulting tractograms reflecting the high orientational order of the human corpus callosum microstructure. The bar graph at the bottom of Fig. 4 displays the myelin levels for both samples. The levels at the body were significantly higher than at the splenium of the corpus callosum, consistent with myelination patterns observed in developing brains assessed with postmortem pathology[2,21].

We also applied SAXS-TT on a human cortex specimen of a 78-year-old female (Supplementary Fig. 3). We chose to image the V1 area (primary visual cortex) because this includes the line of Gennari, a band of myelinated fibers within the cortex that divides the deep and the superficial cortical layers. The contrast between the line of Gennari and the surrounding cortex in tomographic methods such as MRI[22] has been also attributed to iron[23], since iron removal minimizes the contrast[24]. In the selected specimen, the myelinated line was in regions visible to the bare eye (Supplementary Fig. 3a), and could be clearly distinguished in the tomographic SAXS signal reconstruction at the q-values of the myelin peak (Supplementary Fig. 3c). The band was less clearly visible when visualizing the myelin levels based on the myelin-specific signal reconstruction (Supplementary Fig. 3d) due to the significantly lower myelin level values of the line compared to the subcortical white matter. This finding is expected given the advanced age of the donor, which contributes to decreased myelin levels at the line of Gennari[25]. Moreover, the

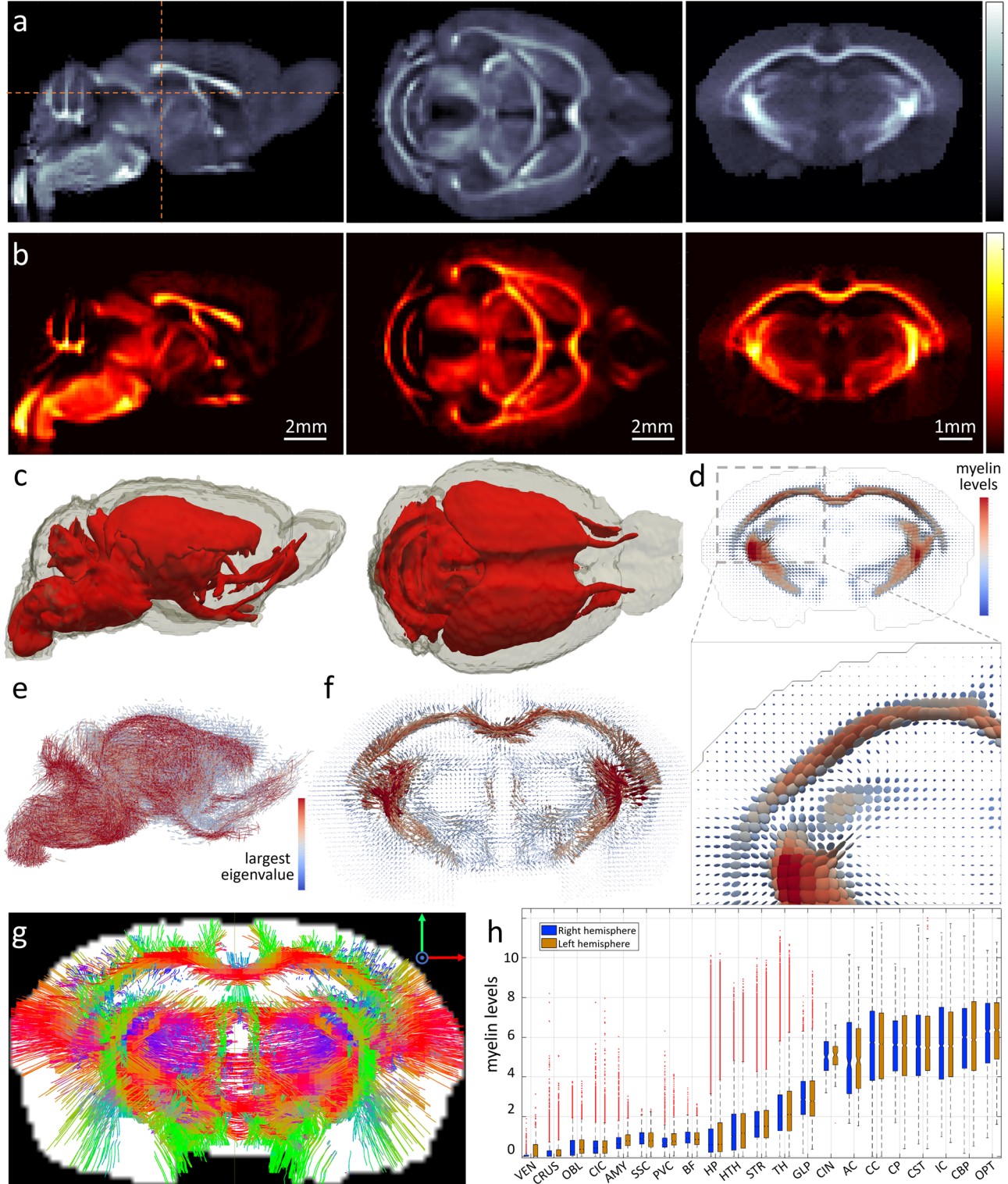

outer cortical layers contained almost no myelin and are barely visible in the myelin map, whereas the deeper cortical layers were more myelinated. These observations are also corroborated by the quantitative values shown in the line plots of Supplementary Fig. 3e: myelin levels decrease sharply going from white matter to the cortex, increase minimally at the position of the line of Gennari, and then decrease to almost zero at the outer cortical layers, in line with the trend observed qualitatively by myelin histology[24–26].

Overall, the myelin specificity of the SAXS-TT method allows quantitatively investigating the myelination within and across human brain specimens, including fine myelinated structures.

**Comparison with myelin histology and MRI**. Myelin levels were validated using myelin histology: we compared the SAXS-TT-derived myelin levels for the C57BL/6 mouse brain with bright-field images of 320 consecutive Luxol-fast-blue-stained 10-µm-

**Fig. 2 SAXS-TT results for mouse brain. a** Sagittal, axial, and coronal virtual slices of tensor-reconstructed reciprocal space-map intensity at myelin peak $q$-values. Grayscale colormap was used for the unspecific scattering signal, akin to unspecific tomographic contrasts such as MRI or CT. **b** Same virtual slices depicting the tensor-reconstructed myelin-specific signal. "Hot" colormap was used for the myelin-specific signal, akin to molecular imaging outcomes, e.g. using fluorescent tags. **c** 3D myelin distribution map of highly myelinated areas (red). **d** Coronal slice of the 3D fODF map, with tensors represented by ellipsoids, colored by tensor trace. **e** Side-view of 3D map of fiber orientations, represented by lines. Color and orientation correspond to the largest tensor eigenvalue. **f** Same coronal slice as in **a**, **b**, **d** from map in **e**. **g** Tractogram of same section based on main fiber orientation. **h** Box plots displaying myelin levels for different regions of the mouse brain, for right and left hemisphere, segmented using the DSURQE mouse brain atlas[48]. Notches represent median values, boxplot upper/lower ends represent 25/75 percentiles, and whiskers extend to ±2.7 standard deviations from the mean. Outliers are in red dots. All white matter regions (CIN, AC, CC, CP, CST, IC, CBP, OPT) displayed statistically similar myelin levels between left and right hemispheres using unpaired two-tailed $t$-test (no multiple comparison correction). [VEN: lateral ventricles, $n_{right}/n_{left} = 319/334$, $p$ value: $4 \times 10^{-9}$; CRUS: crus 1&2, cerebellar lobule 7 (gray matter), $n_{right}/n_{left} = 7844/8591$, $p$ value: $2 \times 10^{-6}$; OBGL: olfactory bulb, granule layer (gray matter), $n_{right}/n_{left} = 5846/5889$, $p$ value: 0.02; CIC: cingulate cortex, $n_{right}/n_{left} = 6273/6004$, $p$ value: $4 \times 10^{-3}$; AMY: amygdala, $n_{right}/n_{left} = 11,191/12,069$, $p$ value: 0; SSC: secondary somatosensory cortex, $n_{right}/n_{left} = 4381/4318$, $p$ value: $3 \times 10^{-9}$; PVC: primary visual cortex, $n_{right}/n_{left} = 887/938$, $p$ value: 0.06; BF: barrel field, $n_{right}/n_{left} = 5695/6516$, $p$ value: $2 \times 10^{-14}$; HP: hippocampus, $n_{right}/n_{left} = 29368/29811$, $p$ value: $10^{-25}$; HTH: hypothalamus, $n_{right}/n_{left} = 13,423/13,413$, $p$ value: $5 \times 10^{-14}$; STR: striatum, $n_{right}/n_{left} = 28218/28766$, $p$ value: $3 \times 10^{-13}$; TH: thalamus, $n_{right}/n_{left} = 22,630/22,812$, $p$ value: $10^{-20}$; GLP: globus pallidus, $n_{right}/n_{left} = 3456/3723$, $p$ value: 0.28; CIN: cingulum, $n_{right}/n_{left} = 162/114$, $p$ value: 0.58; AC: anterior commissure, $n_{right}/n_{left} = 360/426$, $p$ value: 0.43; CC: corpus callosum, $n_{right}/n_{left} = 15,467/15,441$, $p$ value: 0.81; CP: cerebral peduncle, $n_{right}/n_{left} = 2115/2178$, $p$ value: 0.17; CST: cerebrospinal tract, $n_{right}/n_{left} = 2081/1977$, $p$ value: 0.54; IC: internal capsule, $n_{right}/n_{left} = 3304/3221$, $p$ value: 0.84; CBP: cerebellar peduncle, $n_{right}/n_{left} = 3513/3748$, $p$ value: 0.21; OPT: optic tract, $n_{right}/n_{left} = 1862/2056$, $p$ value: 0.70]. Source data are provided as a Source Data file.

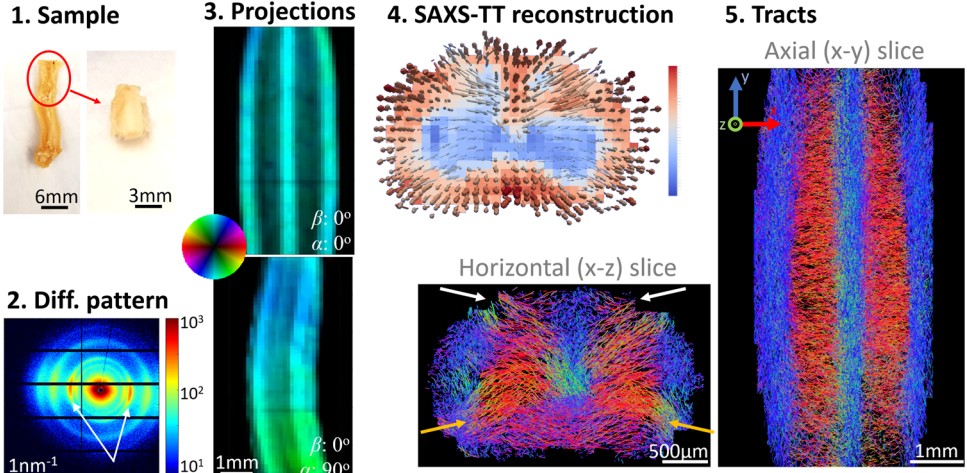

**Fig. 3 SAXS-TT on mouse spinal cord. 1** The cervical part of the fixed spinal cord was scanned and **2** produced strong myelin peaks in the diffraction patterns (white arrows). **3** Each 2D scan resulted in a projection depicting SAXS myelin peak intensity and 2D orientation of the myelinated axons. **4** Tensor-tomographic reconstruction of all projections provided 3D fiber orientations per voxel, here visualized by a vector map for a virtual section. Vectors are colored based on myelin levels in the corresponding voxel. **5** A probabilistic tractography algorithm enabled generating neuronal tracts for the specimen, an axial and a horizontal section of which are shown. Arrows point to anterior (orange) and posterior (white) horns.

thick histological sections covering the same brain's left hemisphere (Fig. 5a). Histology and SAXS-TT results were also compared with MRI data of the same brain (Fig. 5a and Supplementary Fig. 4) using a variety of myelin-sensitive contrasts based on (i) magnetization transfer—MT[27]: MT ratio (MTR) and MT saturation (MT$_{sat}$), (ii) relaxometry: $T_1$ and $T_2$ relaxation times, and (iii) dMRI, including diffusion tensor/kurtosis[28] imaging (DTI/DKI)-derived metrics:[29] fractional anisotropy, radial diffusivity, radial kurtosis, and axonal water fraction (AWF).

Compared to histology, SAXS-TT-derived myelin levels showed the highest linear (Pearson-$r$) and monotonic (Spearman-$\rho$) correlations ($r_{SAXS-TT} = 0.85$, $\rho_{SAXS-TT} = 0.86$). Among the MR metrics, MT parameters displayed the highest correlations with myelin content ($r_{MTsat} = 0.72$, $r_{MTR} = 0.47$, $\rho_{MTsat} = 0.79$, $\rho_{MTR} = 0.72$). AWF showed the highest correlations among the dMRI parameters ($r_{AWF} = 0.52$, $\rho_{AWF} = 0.56$). Spearman correlations were higher than Pearson for most MRI metrics, explained by their monotonic though nonlinear relationships with myelin content, likely because MRI methods probe myelin indirectly through properties of water. Correlation of MRI

metrics with SAXS-TT (Fig. 5a, rows 2 and 4) yielded almost identical coefficients as with histology (Fig. 5a, rows 1 and 3), further supported by the high linear correlations between the coefficients of MRI metrics against both SAXS-TT and histology (Fig. 5b, c), with $r=\sim1$ and the regression lines almost on the identity line. This suggests SAXS-TT's potential as the reference myelin imaging method, with the additional benefits of no extra sample preparation, its nanostructural specificity, and its non-destructive nature.

SAXS-TT-derived axon orientations for a mouse brain (Fig. 2) and a mouse spinal cord (Fig. 3) were compared to those obtained on the same samples from ex vivo diffusion MRI (Fig. 5d–i). The main axon orientations from the two methods are shown in co-registered datasets (Fig. 5d, e, g, h). Co-linearity between orientations from the two methods for all brain voxels is assessed by the dot product of their unit vectors (Fig. 5f, i). For both samples, the dot product distribution is heavily skewed towards 1, indicating that orientations are in good agreement, consistent with previous comparisons between X-ray scattering- and dMRI-derived orientations on thin mouse brain vibratome sections[14].

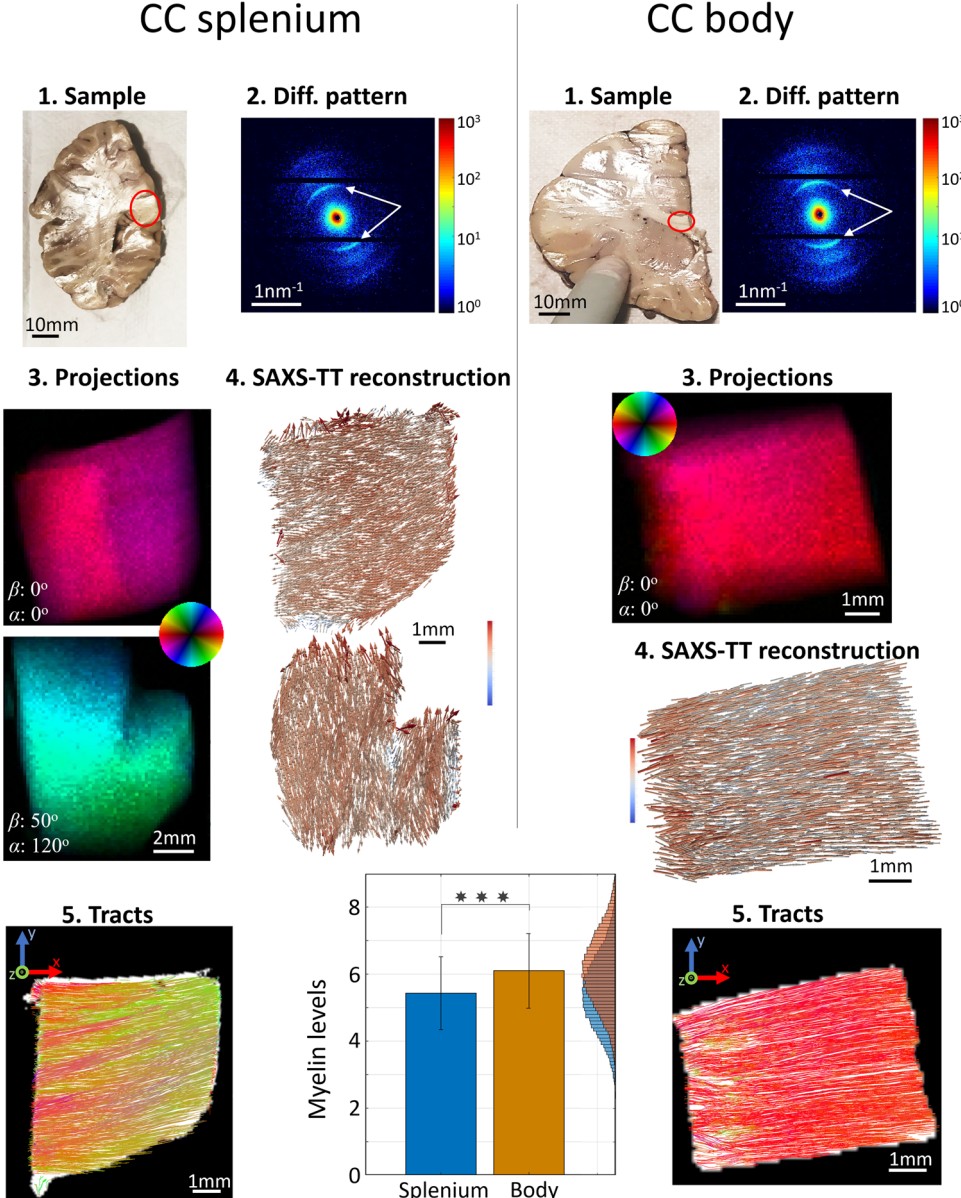

**Fig. 4 SAXS-TT on corpus callosum (CC) splenium and body specimens of a 2-year-old female. 1** Samples, in red circle, were raster scanned and **2** produced strong myelin peaks in their diffraction patterns (white arrows). **3** Each 2D scan resulted in a projection depicting SAXS myelin peak intensity and 2D orientation of the myelinated axons. **4** Tensor-tomographic reconstruction of all projections provided 3D fiber orientations per voxel, here visualized by vector maps for virtual sections. Vectors are colored based on myelin levels in the corresponding voxel. **5** Tractography algorithms enabled generating neuronal tracts for the two specimens, representative sections of which are shown here. The bar graph at the bottom of the figure shows the myelin levels—and respective standard deviations—of the specimens, with the body displaying significantly higher values. Histogram of the values with the same colors is shown to the right of the figure. Significance $p$ values of the unpaired two-tailed $t$-test and the two-sample Kolmogorov–Smirnov test is 0. Splenium contained $n = 70,007$ voxels, and body $n = 51,096$ voxels. Source data are provided as a Source Data file.

Orientations also showed good correspondence with those from 3D histology, using CLARITY[30] with anti-neurofilament antibody staining (Supplementary Fig. 5).

**Differentiating central from peripheral myelin**. To highlight the method's nanostructural specificity and ability for multiplexed analysis, we simultaneously studied central and peripheral nervous system myelin, which are produced by oligodendrocytes and Schwann cells, respectively (Fig. 6a), and associated with distinct myelin pathologies such as multiple sclerosis (CNS) or Guillain-Barré Syndrome (PNS). We exploited their difference with regard to layer periodicity[31] ($d_{CNS} = 15–17$ nm, $d_{PNS} = 19–20$ nm), and studied a mouse brain with cranial nerves not severed during brain

extraction from the skull. The distinct CNS and PNS myelin signals were isolated (Fig. 6b) and mapped for each projection (Fig. 6c). The results show that most cranial/PNS nerves (green) have been severed close to their sprouting site from CNS (magenta) (white arrows), except the proximal part of the trigeminal nerves (orange arrows). The anatomic complementarity of the structures becomes evident when combining the tomographic CNS and PNS reconstructions (Fig. 6d), with the trigeminal nerves fitting to the CNS trigeminal nuclei with a ball-socket fit. Although the trigeminal CNS/PNS transition zone, important in various pathologies[32], has been previously described histologically[33], it is uniquely represented here in 3D due to the combination of SAXS-TT nanostructural specificity, tomographic capabilities, and multiplexed analysis.

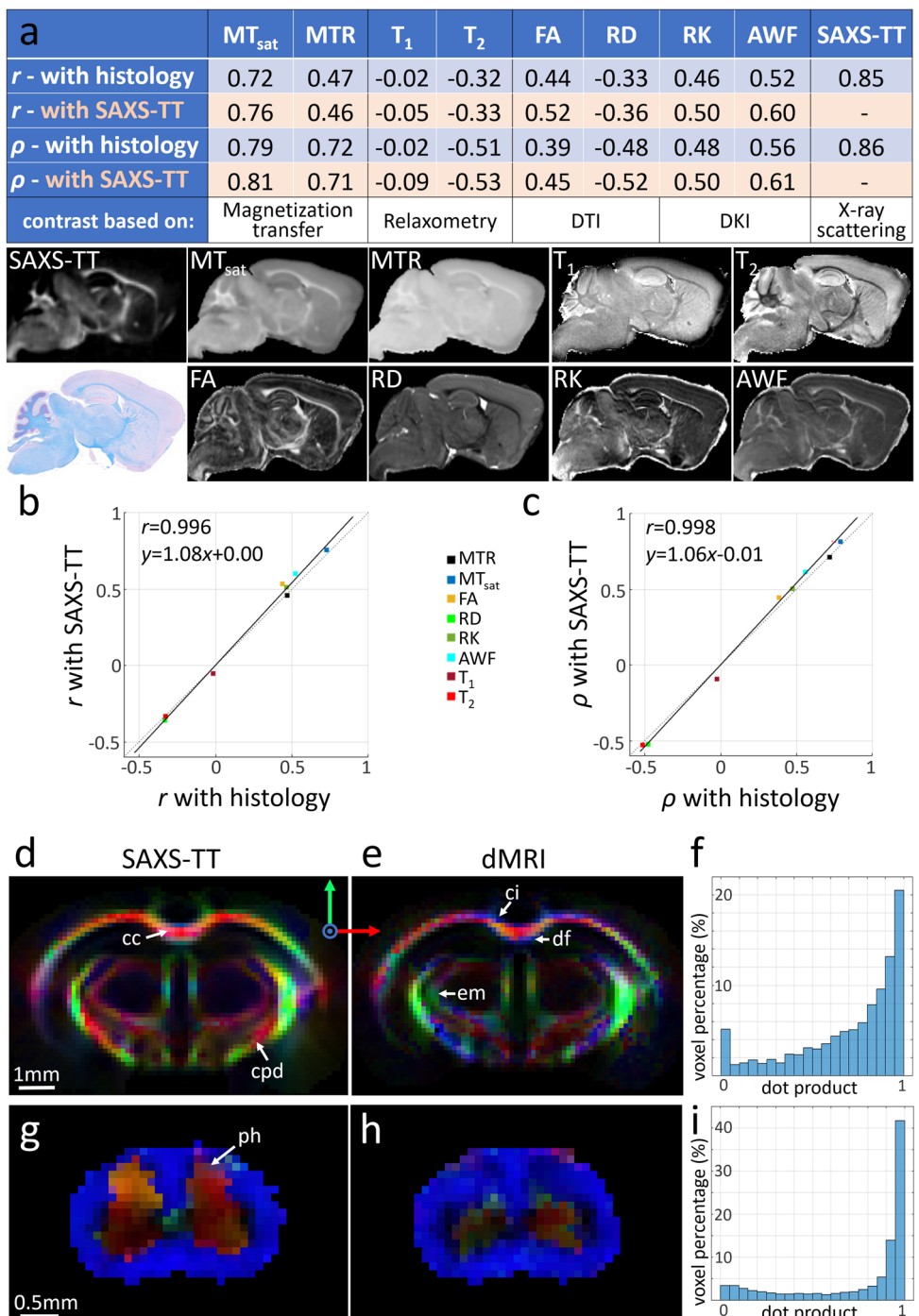

**Fig. 5 Comparing SAXS-TT outcomes to histology and MRI. a** Voxel-wise Pearson ($r$) and Spearman ($\rho$) correlations against histology- and SAXS-TT-derived myelin levels. All $p$ values are 0, correlations over $n = 361,042$ voxels. Figures below table depict the same sagittal slice of the compared parameter maps. MT$_{sat}$ magnetization transfer saturation, MTR magnetization transfer ratio, FA fractional anisotropy, RD radial diffusivity, RK radial kurtosis, AWF axonal water fraction, DTI diffusion tensor imaging, DKI diffusion kurtosis imaging. **b** First against second row of table in **a**, identity line (dotted), linear fit (black line) with indicated equation, and a Pearson correlation of $r = 0.996$ ($p = 1.1 \times 10^{-7}$). **c** Third against fourth row of table in **a**, identity line (dotted), linear fit (black line) with indicated equation, and a Pearson correlation of $r = 0.998$ ($p = 2.5 \times 10^{-8}$). **d, e, g, h** Coronal mouse brain (**d, e**) and horizontal mouse spinal cord (**g, h**) sections depicting SAXS-TT- and dMRI-derived fiber orientations, intensity weighted by SAXS-TT myelin levels, orientations interpreted by red-green-blue arrows. cc corpus callosum, cpd cerebral peduncle, ci cingulum, df dorsal fornix, em external medullary lamina of thalamus, ph posterior horn. **f, i** Dot product between SAXS-TT- and dMRI-derived fiber orientations for brain (**f**) and spinal cord (**i**), where dot-product value of 1 denotes perfect collinearity, and 0 a 90° discrepancy. Source data are provided as a Source Data file.

**Quantitative study of dysmyelination model.** To demonstrate sensitivity to demyelination and dysmyelination pathologies that induces alterations in myelin levels and/or structure with severe neurological implications, we used the shiverer model, which lacks functional compact CNS myelination[34]. Whole-brain SAXS-TT scanning (Fig. 7a) and subsequent myelin-specific signal isolation (Fig. 7b) revealed strong myelin signals for the control brain, and myelin signals of low yet measurable intensity for the shiverer

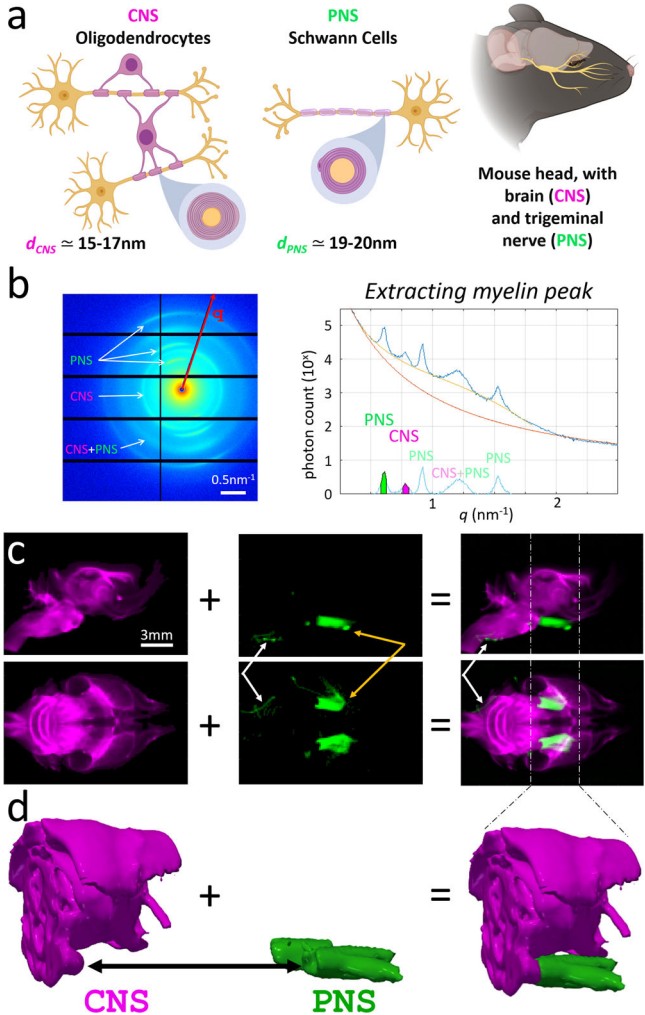

**Fig. 6 Central and peripheral nervous system (CNS and PNS) myelin.**
**a** Oligodendrocyte- and Schwann cell-produced myelin in CNS and PNS, respectively. Trigeminal nerve shown in yellow in the mouse head scheme. Sketches created with BioRender.com. **b** Extracting the myelin-specific signal from intensity vs. $q$ plots, from a scattering pattern including both CNS and PNS peaks (arrows). Top curves: blue → raw data, orange → first fit, yellow → second fit. Bottom curve: myelin signal. Colored areas: identified CNS and PNS peaks used for further analysis. **c** CNS (magenta) and PNS (green) mapping in side- and top-projections, and combined image. Arrows: white → severed brainstem PNS nerves, orange → proximal trigeminal nerves. **d** CNS and PNS 3D myelin maps, showing the ball-socket CNS-PNS fit at the trigeminal nucleus.

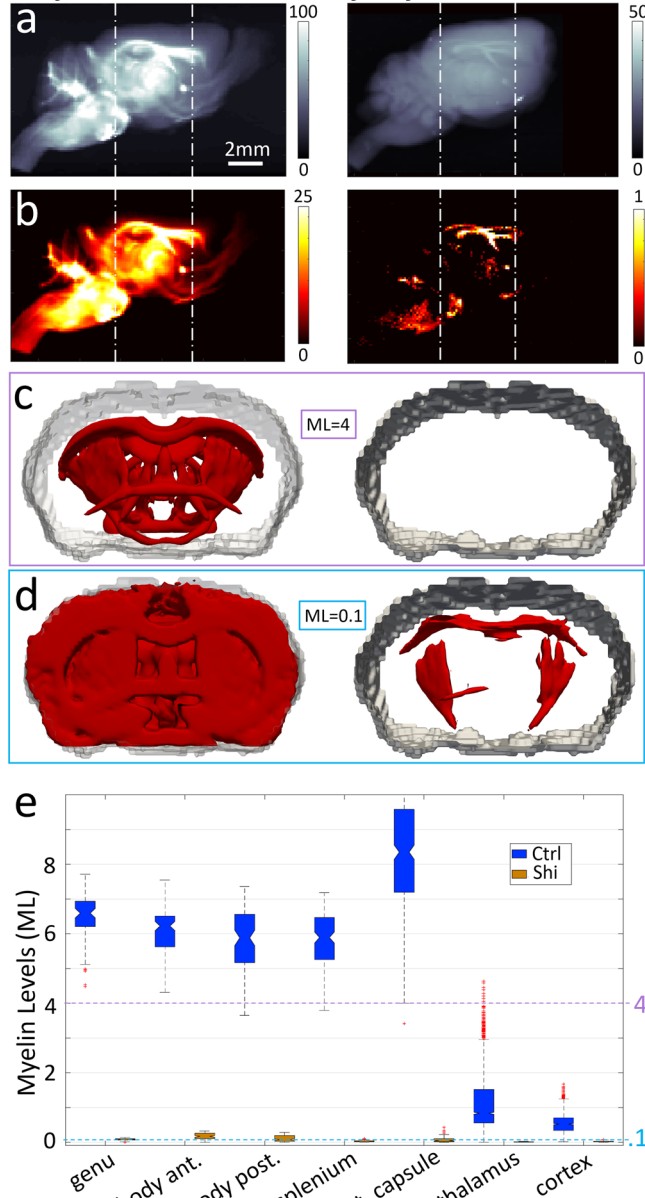

**Fig. 7 Control vs. dysmyelinated (shiverer) mouse brain. a** Side projection of mean scattering intensity at myelin peak. **b** Same projection, with myelin-specific signal isolated. Color bars: photon-count. **c, d** 3D myelin maps at different myelin level (ML) thresholds for the region between dotted lines in **a, b**. ML units: scattered photons/voxel; ML = 4: highly myelinated areas in control mouse appear, no area in shiverer; ML = 0.1: white matter regions just appear in shiverer vs. almost entire brain for control. **e** Myelin levels for control (Ctrl-blue) and shiverer (Shi-orange) brains, for selected areas. ROI voxel numbers for control/shiverer: genu: 71/80, body anterior: 105/94, body posterior: 99/93, splenium: 136/63, internal capsule: 337/119. thalamus: 2740/210, cortex: 1018/403. White matter ROIs were eroded with a 3 × 3 × 3 box kernel to avoid partial volume effects. Box plots: notch → median, box → 25-75 percentiles, whiskers → ±2.7 standard deviations from mean, red crosses → outliers. Source data are provided as a Source Data file.

brain—note the difference in colorbar values. The yielded 3D myelin maps (Fig. 7c, d) enabled quantitative regional assessment of myelin levels (Fig. 7e). For the control brain, highest myelin levels were observed in the internal capsule, with similar myelin levels for corpus callosum subregions, and much lower yet clearly detectable myelination in the cortex. In contrast, myelin levels in the shiverer mouse were very low across all white matter regions, typically ~20–30 times lower than in control mouse white matter, and even ~2–3 times lower than the control mouse cortex, while myelin was barely detectable in the thalamus or cortex. In addition, shiverer myelin exhibited a slightly larger period between layers than in controls ($d_{CNS,shiv} \sim 18.5$ nm vs. $d_{CNS,ctrl} \sim 17$ nm, cf. Supplementary Fig. 6), consistent with the well-reported altered myelin conformation in shiverer CNS observed by electron microscopy[34,35]. Overall, tensor-tomographic reconstruction and

quantification of myelin-specific signals from control and dysmyelinated nervous tissue, highlights SAXS-TT's specificity for probing myelin levels and structure and reveals its potential for characterizing diseases involving myelin alterations.

## Discussion

We have presented a non-destructive framework for nanostructure-specific imaging of myelin, myelin-sheath integrity, and axon orientations in macroscopic samples, demonstrated for mouse and human tissue samples. The specificity and sensitivity to nanostructure periodicity was showcased by selectively imaging CNS and PNS myelin in a multiplexed manner. Furthermore, we illustrated SAXS-TT's sensitivity to myelin levels and integrity within and across samples, also in cases of altered myelination, using a dysmyelination model.

Comparison with histology using myelin stains, the current gold standard, revealed the accuracy of SAXS-TT-based myelin levels. Also, high similarity was found between histology and SAXS-TT when compared to a number of myelin-sensitive MRI metrics. These findings suggest that SAXS-TT could serve as a non-destructive reference method for assessing myelin levels in intact samples. It can thereby avoid the tedious and artifacts-inducing sample preparation, sectioning and staining steps of histologic processing, while leaving the sample intact for further investigations. At the same time, it can provide specificity to myelin, unavailable with macroscopic imaging techniques.

High correlations were found between SAXS-TT- and dMRI-derived axon orientations. Discrepancies were observed in some voxels, highlighted in Fig. 5. These can be attributed to small voxel-level registration errors, to voxels with almost isotropic orientations, and to SAXS-TT and dMRI measuring very different biophysical phenomena and micro-/nanostructural characteristics: SAXS-TT is sensitive to myelinated axons only, while dMRI signals comprise contributions from multiple structures and intra- or extra-axonal compartments[36]. Fiber orientations derived from microscopic analysis using tissue clearing and immunostaining were in qualitative agreement with SAXS-TT orientations. It should be noted that orientations from tissue clearing and immunostaining were challenging to quantitatively compare across samples because of nonlinear tissue distortions, anisotropic voxels, inhomogeneous antibody penetration, high axonal density regions where signal appears homogeneous at optical microscopy resolution, etc.[37,38]. Moreover, neurofilament staining did not stain myelinated axons only, but all axons. Myelin-specific stains, e.g., targeting myelin basic protein (MBP) or myelin proteolipid protein (PLP), would allow such quantification, and will be used in future studies aiming at direct quantitative comparisons, though they also have limitations, e.g. MBP cannot be used on models such as shiverer mice that lack the protein. All these further highlight the need for a reference method for myelinated axon orientations in macroscopic samples, a role we suggest could be assumed by the presented SAXS-TT-based method.

Limitations of the method include the radiation dose imparted by X-rays, which may result in significant damage on biological tissue[39]. However, the imparted dose and dose rate in the presented experiments seemed to be in a range that did not affect the tissue morphology at the investigated length scales: first, scan of the same sample projection at the beginning and the end of the X-ray experiment (Supplementary Fig. 7a) showed identical X-ray scattering signals. Second, MRI scans performed with the same parameters before and after SAXS-TT (Supplementary Fig. 7b) did not reveal detectable contrast differences—minor anatomical discrepancies are due to minimal changes in slice position. It should be additionally noted that SAXS-TT scanning is fully compatible with further processing of the sample, for instance for 2D and 3D histology, staining, and imaging as presented above. Moreover, synchrotron X-ray methods become increasingly compatible with in vivo investigations[40], a direction we aim to pursue in future studies.

Also, in the experiments discussed, specimens were extracted from surrounding bones to avoid interference from dense structures, though myelin signals can still be detected from within the skull (Supplementary Fig. 8). Moreover, the rank-2 tensor fODF representation used here cannot resolve complex within-voxel fiber configurations such as crossing fibers. This could be overcome by increasing the resolution or by using higher-order fODF representations, e.g., based on spherical harmonics[15,17]. Further, in control and shiverer nervous tissue experiments a single rotation axis was used, which did not result in an apparent loss of information, though future validation studies should investigate optimal acquisition strategies.

A challenge of SAXS-TT experiments is the long scan times needed (cf. Table in Methods). Scan duration depends on the target resolution; scan duration is —approximately— inversely proportional to the cube of the voxel dimension. For instance, decreasing the (isotropic) voxel size from 200 to 100 µm —while keeping the exposure time the same— will induce an ~8-fold increase in scan time. The multi-hour scans needed to cover larger samples with adequate resolution might strain the sample and system stability. To that effect, the beam intensity should be tracked and taken into account, as explained in the "Methods" section. Also, we have observed that higher agarose gel concentrations (>1% w/v) result in gel evaporation and the formation of bubbles during the scans, and should thus be avoided. Moreover, long SAXS-TT acquisition times require allocation of long synchrotron beamtimes for studies with multiple samples. Yet, beamlines are increasingly capable of accommodating multi-sample experiments, due to the implementation of on-the-fly scanning, combined with high-sensitivity detectors and higher fluxes available in modern synchrotrons.

When it comes to comparisons within and across samples, specific care has to be taken when performing quantitative analyses. Within a single sample or across samples scanned in the same beamtime, the fluctuations of the incoming X-ray beam during the experiment have to be taken into account. This can be achieved either by a dedicated sensor constantly measuring the incoming beam, or by the off-sample intensity measured by the beamstop in every line of scanning. For comparisons between samples across different beamtimes or beamlines, quantification of the SAXS intensity is needed, which is possible by using SAXS calibration standards[41]. This approach will also enable benchmarking the intra- and inter-sample variability of the method across multiple samples, beamtimes, and beamlines, extending the variability analyses presented in Figs. 2 and 4.

The detailed information on myelin levels, integrity, and axon orientations provided by SAXS-TT could be helpful for mechanistic studies, e.g., on nervous system maturation during brain development or aging-related changes, to phenotype pathologies affecting myelin levels and integrity, or to characterize the response to therapeutic interventions. Moreover, it can be used for validating non-invasive imaging readouts targeting myelin, axon orientation, or nervous tissue microstructure[36].

Future studies will explore these ideas, and aim at enhancing the method, e.g. by wide-angle X-ray scattering[42] or X-ray fluorescence measurements[43], thus providing information from other nervous tissue components exhibiting nano-periodicities such as lipid membranes, tissue water, or amyloid fibrils[44–46], or enable elemental mapping[47], respectively. Extension to other ordered macromolecules or nanostructures is straightforward, also in a multiplexed manner, enabling a comprehensive quantitative tomographic tissue characterization in normal or altered state.

## Methods

**Samples and sample preparation.** A 5-month-old C57BL/6 female mouse was imaged in the multimodal (MRI, SAXS-TT, 2D and 3D histology) mouse brain

study. Animal was group-housed in individually ventilated cages in a temperature-, humidity-, and light-controlled environment (22 °C, 45–50%, 12 h light/dark cycle), and had access to food and water ad libitum. The terminally anesthetized mouse (pentobarbital 100 mg/kg) was transcardially perfused with 1% phosphate-buffered saline (PBS) and 4% paraformaldehyde (PFA), and its brain immediately extracted and stored in 4% PFA for 48 h at 4 °C. Subsequently it was transferred to 1% PBS, and kept at 4 °C for 2 weeks until the MRI experiments. After the MRI scanning, it was kept again in PBS at 4 °C for another 2 weeks, until the SAXS-TT experiments. Following the SAXS-TT experiments, it was MRI-scanned again to test possible effects from SAXS-TT scanning, and no alteration of the MRI signal maps was observed. After two months in 1% PBS at 4 °C, the brain was cut in half at the mid-sagittal plane. The left hemisphere was sent for histology sectioning and myelin staining. The right hemisphere was taken for passive CLARITY[30] with antibody staining. Experiments were under the animal license ZH242/14 of the Animal Imaging Center of ETH Zurich/ University of Zurich.

For the mouse spinal cord experiment, a 50-day-old Rag2$^{-/-}$ male mouse was used. Mouse was housed in 12:12 light:dark light cycles at room temperatures ranging between 20 and 26 °C and humidities between 40 and 60%. Every solid-bottomed, contact-bedded cage housed one to five adult mice, with food and water access ad libitum. A terminally anesthetized animal (pentobarbital 100 mg/kg) was transcardially perfused using pressure-controlled Perfusion ONE system (Leica) with 10% sucrose solution followed by 4% phosphate-buffered PFA. Pressure throughout the perfusion procedure was maintained at 150 mmHg. The cervical part of the cord was excised and stored in 1% PBS at 4 °C for 24 h until the MRI experiments. After MRI scanning, it was placed back in 1% PBS at 4 °C for 3 days until the SAXS-TT experiments. Animal study was reviewed and approved by the Johns Hopkins University animal care and use committee protocol number MO16M313.

The human white matter samples (corpus callosum splenium and corpus callosum body) were excised from a formalin-fixed brain of a 24.9 month female subject with no indication of neuropathology, from the Sudden and Unexpected Death in Children (SUDC) biobank of NYU Langone. The brain had been preserved in formalin for 40 days, and then washed for 3 days in 1% PBS solution, renewed every day. Experiments were in accordance with the NYU Langone Institutional Review Board decision for study i14-01061, with the tissue having been donated with written consent of the parents and in the absence of compensation.

The human primary visual cortex sample was excised from a formalin-fixed brain of a 78-year-old female with no pathological finding in the cortex, from the tissue bank of Stanford's Alzheimer's Disease Research Center (ADRC), with the donor's informed consent. The brain was preserved in 1% PBS + 0.02% sodium azide solution for approximately 3 years. The study was in accordance with Stanford University Human Subjects Research Institutional Review Board, protocol # 33727.

The Rag2$^{-/-}$ and Rag2$^{-/-}$sh$^{-/-}$ mouse brains were from 50-day-old sex- and age-matched mice, retrieved, and stored with the same procedure and under the same license as the spinal cord.

**SAXS-TT experiments, reconstruction, and myelin signal extraction**. The C57BL/6 mouse brain and the human white matter specimens were scanned in the cSAXS beamline of the Swiss Light Source (SLS) synchrotron (Supplementary Fig. 1a) in the Paul Scherrer Institute (PSI), Villigen, Switzerland. The Rag2$^{-/-}$ mouse spinal cord and Rag2$^{-/-}$ and Rag2$^{-/-}$sh$^{-/-}$ mouse brains were scanned in the LiX beamline of the National Synchrotron Light Source-II synchrotron (Supplementary Fig. 1b), in the Brookhaven National Laboratory (BNL) (Brookhaven, NY, USA). The human cortex specimen was scanned in the 4-2 beamline of the Stanford Synchrotron Radiation Lightsource (SSRL) (Supplementary Fig. 1c) in the SLAC National Accelerator Laboratory (CA, USA). All samples were embedded in 1% agarose gel, with 1% PBS as a buffer, and placed within 10-mm-diameter (3 mm for the mouse spinal cord, 20 mm for the cortex specimen) Kapton tubes (Goodfellow Inc., UK) for scanning. Experiment details and radiation dose for all samples are presented in Table 1.

Before tomographically reconstructing the SAXS signals, these were corrected for photon absorption in the sample by dividing them by the transmitted beam intensity as recorded by the photodiode in front of the SAXS detector. Further, in order to directly compare the Rag2$^{-/-}$ and Rag2$^{-/-}$sh$^{-/-}$ myelin levels, the SAXS intensities were normalized with the intensity of the incoming beam, as recorded upstream of the sample. The IRTT algorithm[18] was used for SAXS-TT reconstructions, with the number of iterations used for each sample indicated in Table 1.

In order to extract the myelin-specific signal for every segment in each scattering pattern (cf. Fig. 1a, b), a three-step procedure was followed, as visualized in Fig. 6b, Supplementary Fig. 6, and Supplementary Movie 4 for all diffraction patterns along a line scan across a mouse brain. The first two steps comprise a background fit. Initially, an exponential curve of type $I = \frac{a}{(q+b)^c} + d$ was fitted to the signal at q-values above and below the observed myelin peaks: $q < 0.4$ nm$^{-1}$ and $q > 2$ nm$^{-1}$ to capture the main background trend ($I$: signal intensity; $q$: reciprocal space coordinate; $a$, $b$, $c$, $d$: free regression parameters). After subtracting the fitted curve, a fourth-degree polynomial was fitted to the remaining signal, at the positions where no peak is observed or expected: $0.4$ nm$^{-1} < q < 0.48$ nm$^{-1}$, $1.02$ nm$^{-1} < q < 1.05$ nm$^{-1}$, $1.35$ nm$^{-1} < q < 1.49$ nm$^{-1}$, and $1.601$, nm$^{-1} < q < 1.75$ nm$^{-1}$. Following subtraction of the polynomial, only the myelin-specific signal, if any, remained (Fig. 6b, Supplementary Fig. 6 and Supplementary Movie 4). If a CNS myelin peak was detected in the q-range $0.72$ nm$^{-1} < q < 0.85$ nm$^{-1}$, the area under the peak was quantified. Similarly, if a PNS peak was detected in the range $0.58$ nm$^{-1} < q < 0.65$ nm$^{-1}$, its area was quantified. For the shiverer brain, myelin-specific signal area was quantified at the q-values corresponding to its altered CNS myelin periodicity $0.65$ nm$^{-1} < q < 0.71$ nm$^{-1}$. The brain ROIs presented in the quantitative myelin level analysis in Fig. 2h were defined using the DSURQE mouse atlas[48].

**MRI experiments and parameter estimation**. All samples were embedded in a perfluorocarbon solution (Fomblin®) during MRI scanning.

*C57BL/6 mouse brain*. MRI scanning of the C57BL/6 mouse brain was performed ex vivo on a 9.4T Bruker BioSpec 94/30 scanner (Bruker Biospin GmbH, Ettlingen, Germany) of the Animal Imaging Center of ETH Zurich and University of Zurich, using a semi-cylindrical 4-channel cryogenic surface coil (CryoProbe, Bruker

## Table 1 Experimental and reconstruction details for SAXS-TT scans.

| | C57BL/6 brain (mouse) | Spinal cord (mouse) | V1 cortex (human) | CC splenium/ body (human) | Rag2$^{-/-}$(sh$^{-/-}$) brains (mouse) |
|---|---|---|---|---|---|
| Beamline, synchrotron | cSAXS, SLS | LiX, NSLS-II | 4-2, SSRL | cSAXS, SLS | LiX, NSLS-II |
| Tube diameter (mm) | 10 | 3 | 20[a] | 10 | 10 |
| Beam energy (keV) | 16.3 | 15.7 | 17 | 16.3 | 16.9 |
| Beam size (μm$^2$) | 150 × 150 | 100 × 100 | 250 × 250 | 150 × 150 | 120 × 120 |
| Motor step size (μm) | 150 | 100 | 250 | 150 | 120 |
| Field of view (pixels) | 69 × 105 | 38 × 61 | 85 × 90 | 70 × 82 | 91 × 41 |
| Field of view, size (mm$^2$) | 10.35 × 15.75 | 3.8 × 6.1 | 21 × 22.5[a] | 10.5 × 12.3 | 10.92 × 4.92 |
| No. of projections | 267 | 60 | 151 | 254 | 181 |
| Rotation axis tilt angles (°) | 0, 10, 15, 20, 30, 40, 45, 50 | 0 | 0 | 0, 10, 20, 30, 40, 50 | 0 |
| Nr of projections/tilt | 67, 34, 32, 32, 30, 26, 24, 22 | 60 | 151 | 66, 56, 50, 44, 38 | 181 |
| Exposure time/frame (ms) | 125 | 300 | 120 | 80 | 100 |
| Total exposure time (h) | 67.1 | 11.6 | 38.5[a] | 25.3 | 18.8 |
| Total scan time (h) | 87.2 | 17.4 | 61[a] | 36.8 | 24.8 |
| Dose[b] (Gy) | 2.33 × 10$^4$ | 9.96 × 10$^4$ | 1.1 × 10$^4$ | 1.38 × 10$^4$ | 2.48 × 10$^4$ |
| Reconstruction iterations | 10,000 | 2,000 | 5,000 | 5,000 | 5,000 |

[a] These values include scanning of two more specimens, not incorporated in the present study, that were scanned together with the V1 sample (in the same tube, next to one another, see Supplementary Fig. 1).
[b] Dose in Gy ($= \frac{energy\,(J)}{mass\,(kg)}$) was calculated as $\frac{(photon\,flux) \times (exposure\,time) \times (tissue\,absorbtion) \times (photon\,energy)}{(tissue\,volume) \times (tissue\,density)}$

Biospin GmbH, Fälanden, Switzerland) with dimension $20 \times 27$ mm$^2$ for signal reception and a resonator probe of 72 mm inner diameter for excitation.

For the dMRI scans, diffusion encoding was applied along 200 directions, of which 20 directions for $b = 1$ ms/μm$^2$, 40 for 2 ms/μm$^2$, 60 for 3 ms/μm$^2$, and 80 for 4 ms/μm$^2$, along with 5 $b = 0$ scans for each $b$-value. Echo time (TE) was 42.9 ms, repetition time (TR) was 500 ms, diffusion gradient duration was $\delta = 5.5$ ms, gradient separation was $\Delta = 12.1$ ms, with isotropic voxel size 75 μm, 2 averages, and a segmented 3D SE-EPI (spin-echo echo planar imaging) readout for a $173 \times 126 \times 211$ ($=12.975 \times 9.45 \times 15.825$ mm$^3$) matrix. Total scan time was 51.5 h. Diffusion and kurtosis parameters[29] were estimated using the DESIGNER pipeline[49], which includes removal of noise[50,51] and Gibbs artifacts, correction for inhomogeneities of the $B_1$ field, as well as eddy current correction.

For the MTR scan, a Bruker FLASH (fast low angle shot) sequence was used, with TE = 3.5 ms, TR = 400 ms, 6 averages, isotropic voxel size 150 μm, matrix $113 \times 73 \times 45$ ($=17 \times 11 \times 6.7$ mm$^3$). The MT pulse was applied at an offset of 1500 Hz with a $B_1$-amplitude of 40 μT. Total scan time was 4.8 h. The MTR was calculated as MTR $= (\mathrm{MT_{NOPULSE}} - \mathrm{MT_{PULSE}})/\mathrm{MT_{NOPULSE}}$.

The quantitative MT scans were performed according to the multi-parameter mapping method[27,52] using proton-density-weighted (PD$_W$), magnetization-transfer-weighted (MT$_W$), and T$_1$-weigthed (T$_{1W}$) MGE (multi-gradient echo) scans at 14 TEs from 1.5 to 14.5 ms every 1 ms, TR = 25 ms, flip angle: MT$_W$, PD$_W$ = 6°, T$_{1W}$ = 15°, 20 averages, isotropic voxel size 150 μm, matrix $155 \times 133 \times 59$ ($=23.25 \times 19.95 \times 8.85$ mm$^3$). For the MT$_W$ scan, the MT pulse was applied at an offset of 3000 Hz with a $B_1$-amplitude of 10 μT. Total scan time was 2.5 h. MT$_{sat}$ was calculated from Matlab scripts based on equations in ref.[52].

For the T$_1$ map, a Bruker RAREVTR (Rapid Acquisition with Refocused Echoes and Variable TR) sequence was used, with TE = 7.1 ms, multiple TRs at [100, 200, 400, 800, 1000, 1200, 1400, 1600, 2000, 3000] ms, isotropic voxel size 150 μm, matrix $226 \times 147 \times 110$ ($=17 \times 11 \times 8.25$ mm$^3$). Total scan time was 19.7 h. A mono-exponential fit was used to retrieve the T$_1$ relaxation value for every voxel.

For the T$_2$ map, a Bruker MSME (Multi-Slice Multi-Echo) sequence was used, with TR = 3000 ms, 25 TEs from 8.3 to 207.5 ms every 8.3 ms, isotropic voxel size 150 μm, matrix $226 \times 147 \times 110$ ($=17 \times 11 \times 8.25$ mm$^3$). Total scan time was 10.2 h. A mono-exponential fit was used to retrieve the T$_2$ relaxation value for every voxel.

For the anatomical maps in Supplementary Fig. 7b, used for pre- and post-scan comparison, the following sequences were used: (i) top row: T$_2$-weighted Bruker RARE sequence, first TE: 6.1 ms, effective TE: 48.7 ms, rare factor: 16, TR = 2 s, isotropic voxel size 100 μm, matrix $150 \times 200 \times 80$ ($=15 \times 20 \times 8$ mm$^3$), scan time: 16 mins; (ii) bottom row: T$_1$-weighted Bruker FLASH (fast low angle shot) sequence, TE = 4.5 ms, TR = 250 ms, flip angle: 15°, isotropic voxel size 100 μm, matrix $170 \times 110 \times 67$ ($=17 \times 11 \times 6.7$ mm$^3$), scan time: 33 mins 27 s.

*Rag2*$^{-/-}$ *mouse spinal cord.* Scanning of the Rag2$^{-/-}$ mouse cervical spinal cord was performed ex vivo on the 7T Bruker 70/30 Biospec System of the Preclinical Imaging Center of the NYU Langone, using a quadrature surface receiver. The diffusion scans were performed using a GRASE (GRAdient and Spin Echo) sequence. Diffusion encoding was applied along 170 directions, of which 5 directions for $b$-value = 0.5 ms/μm$^2$, 10 for 1 ms/μm$^2$, 15 for 2.5 ms/μm$^2$, 20 for 5 ms/μm$^2$, 30 for 10 ms/μm$^2$, 40 for 15 ms/μm$^2$, and 50 for 20 ms/μm$^2$, along with 2 $b = 0$ scans for each $b$-value. TE = 39.9 ms, TR was 400 ms, diffusion gradient duration was $\delta = 8.5$ ms, gradient separation was $\Delta = 17$ ms, with isotropic voxel size 100 μm, matrix $128 \times 104 \times 120$ ($=12.8 \times 10.4 \times 12$ mm$^3$); the larger-than-needed field of view covered more samples, not presented here. Total scan time was 12.3 h.

**Tractography.** Tractography for all samples was performed in MRtrix3, using the *tckgen* command. For the C57BL/6 mouse brain and the human white matter SAXS-TT-derived tracts, *tckgen*'s FACT algorithm was used, based on the main fiber orientation. For the spinal cord, *tckgen*'s Tensor_Prob algorithm was used both for SAXS-TT and dMRI datasets. For the implementation of Tensor_Prob on SAXS-TT data, a synthetic dMRI dataset was generated, based on the values of the fODF tensors.

**Myelin histology and imaging.** The left hemisphere of the SAXS-TT- and MRI-scanned C57BL/6 mouse brain was used in the histology experiments to determine myelin levels. The hemisphere was dehydrated through a series of graded ethanol baths, followed by a xylene bath, and placed in a wax bath for an hour at 60 °C. The sample was subsequently sectioned at 340 consecutive 10 μm slices using a Leica rotary microtome (Leica RM2125 RTS, Leica Microsystems AG, Wetzlar, Germany) and each slice was immersed into a Luxol-fast-blue bath at room temperature for 24 h. The paraffin-embedding, sectioning, and staining procedure was performed in Sophistolab AG, Switzerland.

Subsequent imaging was performed at the imaging facilities of ETH Zurich, using the Pannoramic 250 Flash III slide scanner (3DHISTECH Ltd, Hungary) with a ×20 objective, at a resolution of 0.9 μm/pixel. Images of the 320 consecutive slices were taken, while the 20 most lateral ones were rejected due to poor section quality because of the very little tissue content, and slices were registered into a 3D dataset using FIJI. The dataset was down-sampled to 75 μm isotropic voxel size, and the SAXS-TT and MRI datasets were non-linearly registered to it using a Matlab script[53]. Pearson and Spearman correlations were assessed for all voxels of the hemisphere (361,042 voxels) with all $p$ values of reported correlations being 0 due to the large sample size.

**3D histology: clearing, staining, imaging, and deriving orientations.** The right hemisphere of the SAXS-TT- and MRI-scanned C57BL/6 mouse brain was cleared using passive CLARITY[30] in the facilities of Stanford University's Wu-Tsai Neuroscience Institute: the sample was for 2 months in 4% sodium dodecyl sulfate (SDS)/borate clearing buffer (SBC) clearing solution, 1 week in SWITCH off solution (0.5 mM SDS in 1× PBS, 1:500 NF primary antibody), 3.5 days in anti-neurofilament primary antibody (chicken polyclonal anti-neurofilament heavy polypeptide IgY antibody, ab4680, Abcam PLC, UK) SWITCH on solution (1× PBS and 0.2% Triton X-100 (PBS-T), 1:500 NF primary antibody), then PBS-T-washed for 2.5 days, immersed in secondary antibody SWITCH on solution for 11 days, and then 5 days in SWITCH on solution with secondary antibody with 1:500 AF-647 fluorophore (Alexa Fluor® 647 AffiniPure F(ab')$_2$ Fragment Donkey Anti-Chicken IgY (IgG) (H+L), Jackson ImmunoResearch Inc., USA). All SWITCH off/on immersions were at 37 °C, and in parallel with gentle shaking.

Subsequently, the brain was imaged using confocal microscopy at the imaging facilities of ETH Zurich, after being immersed into RapiClear 1.47 (SunJin Lab Co., Taiwan). Confocal microscopy was chosen for imaging because it yielded superior results to multiphoton microscopy (~×10 detector counts) in a comparison session. A Leica TCS SP8 MP multiphoton microscope was used, with a ×10 objective, a field of view of $512 \times 512$, an excitation wavelength of 633 nm, an in-plane resolution of 3 μm, and a through-plane resolution of 6 μm, with the latter matching the size of the instrument's point spread function in this direction. Each separate image was corrected for the instrument's inherent intensity gradients (producing, e.g., lower intensity at the edges and higher in the middle of each frame) using an image of a homogeneously distributed AF-647 fluorophore as a reference for correction. All images in each plane were subsequently stitched together using FIJI.

Fiber orientations were calculated based on structure tensor analysis[54,55] for $60 \times 60 \times 60$ μm$^3$ voxels, and fractional anisotropy was derived based on tensor eigenvalues. The 3D intensity image was also scaled at $60 \times 60 \times 60$ μm$^3$ voxel size, and registered to the SAXS-TT and dMRI space using Matlab[53] with subsequent registration of the orientation dataset using the same transform. For color-encoded fiber orientation images in Supplementary Fig. 5h-i, vectors were scaled with the square of staining intensity, to accentuate the white matter portions of the sample and to suppress autofluorescence, thus facilitating qualitative comparisons with the SAXS-TT and dMRI datasets.

**Reporting summary.** Further information on research design is available in the Nature Research Reporting Summary linked to this article.

## Data availability

Data have been deposited in https://doi.org/10.16907/0cf7e016-9ac1-4546-8eca-bb59c1a61e34. Source data are provided with this paper.

## Code availability

SAXS data handling has been done using the "Base package", "Scanning SAXS package", and "SASTT package" of the cSAXS beamline, Swiss Light Source, found at https://www.psi.ch/en/sls/csaxs/software. IRTT reconstruction code (included in the SASTT package) can also be found at https://zenodo.org/record/1480589#.YBefZ2RKjAy. The DESIGNER pipeline code used for MRI post-processing and parameter extraction is at https://github.com/NYU-DiffusionMRI/DESIGNER. The myelin-specific signal extraction code has been deposited at https://doi.org/10.5281/zenodo.4496831.

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

## Acknowledgements

We would like to thank Giovanna Diletta Ielacqua from ETH Zurich for the C57BL/6 mouse brain extraction from the skull, Markus Vaas from ETH Zurich for assistance in histology slides imaging, Justine Kusch-Wieser from ETH Zurich for assistance in confocal imaging, Silvia Behnke from Sophistolab AG for the histology tissue processing, and Arline Faustin from NYU Langone for cutting the SUDC corpus callosum speci-mens. We thank for constructive discussions with Youssef Zaim Wadghiri, Dina Ramadane, Gregory Lemberskiy, Antonios Papaioannou, and Hong-Hsi Lee from NYU Langone, Foivos Georgiadis from ETH Zurich, Maged Goubran and Amir Pirmoazen from Stanford University, and Andreas Menzel and Ana Diaz from Paul Scherrer Institute. This work was funded by Swiss National Science Foundation (SNSF) grant numbers P2EZP3_168920, P400PM_180773, 200021_178788 by the National Institutes of Health (NIH) award numbers R01 NS088040, R01 AG06112001, S10OD021512, NIGMS T34 6M096958, NIH/NCI 5P30CA016087, NIH/NIBIB P41 EB017183, NIH/NIGMS P41 GM111244, NIH/NIGMS P41GM103393, S10 OD012331 and US Depart-ment of Energy grant number KP1605010 and contract number DE-AC02-76SF00515.

## Author contributions

M.G. conceived the project, planned and performed experiments and analyses, and wrote the manuscript, with contributions from co-authors as indicated. A.S. co-planned all C57BL/6 mouse brain experiments and co-performed the C57BL/6 mouse brain MRI experiments. Z.G. co-performed C57BL/6 mouse brain SAXS-TT and MRI experiments and analysis. M.G.-S. co-performed SAXS-TT experiments in the cSAXS beamline. M.L. assisted with C57BL/6 mouse brain SAXS-TT experiments. C.L. led CLARITY-related experiments. C.L. and J.A.M. co-planned CLARITY-related experiments. A.B. co-performed CNS-PNS-related analysis. J.V. contributed ideas and code in tractography analysis, and co-planned C57BL/6 mouse brain diffusion MRI experiments. B.A.-A. assisted with dMRI analyses. S.K. co-performed experiments in LiX beamline. T.S. co-planned and assisted with human white matter experiments. C.H.L. assisted with the spinal cord, Rag2$^{-/-}$, and Rag2$^{-/-}$sh$^{-/-}$ sample preparation and the spinal cord MRI experiments. P.W. co-planned the Rag2$^{-/-}$ and Rag2$^{-/-}$sh$^{-/-}$ study. S.C. co-performed

experiments in the LiX beamline. P.D. assisted with human cortex experiment planning and execution. G.D. assisted with planning, experiments, and analysis related to magnetization transfer. M.A. assisted with all MRI C57BL/6 mouse brain experiments. V.Z. co-planned and co-performed the diffusion MRI C57BL/6 mouse brain experiments. S.S. co-planned the SAXS-TT and diffusion MRI C57BL/6 mouse brain experiments. I.R. and T.W. co-planned and I.R. co-performed the human cortex experiments. O.B. co-planned SAXS-TT C57BL/6 mouse brain experiments and assisted in the analysis of Rag2$^{-/-}$ and Rag2$^{-/-}$sh$^{-/-}$ samples. J.Z. assisted with the spinal cord, Rag2$^{-/-}$, and Rag2$^{-/-}$sh$^{-/-}$ sample preparation and the spinal cord MRI experiments, and co-planned the Rag2$^{-/-}$ and Rag2$^{-/-}$sh$^{-/-}$ study. D.S.N. co-planned SAXS-TT and MRI experiments on spinal cord, human white matter, Rag2$^{-/-}$, and Rag2$^{-/-}$sh$^{-/-}$ brains, and MRI analyses. M.Z. co-planned CLARITY and human cortex experiments, and the CNS-PNS study and related analysis. E.F. and M.R. co-planned the project, experiments, and analyses. All co-authors critically revised and provided feedback for the manuscript.

## Competing interests

The authors declare no competing interests.
