## [Peer Review File · Nature Communications]

Reviewer #1 (Remarks to the Author):

In the present work, the authors demonstrate a novel x-ray-based technique, SAXS-TT, for specific labeling of central and peripheral myelin sheaths in fixed tissue specimens *ex vivo* that utilizes a synchrotron-derived beam line. They present data on whole mouse brain, mouse spinal cord as well as human brain tissue, and validate their findings using different myelin-susceptible MRI weightings, standard Luxol-fast Blue histochemistry as well as tissue clearing and antibody stainings. Furthermore, they validate their level of myelin detection by using the dysmyelinating shiver mouse. They also exploit differences in CNS and PNS myelin nanostructure and periodicity to differentially display both types of myelin. They also show that their method is able to detect cortical myelin that is approximately 10 fold less dense than white matter myelin, demonstrating its usefulness in the settings of de- and dysmyelination.

Myelin structure and function are increasingly recognized to be affected not only in classical demyelinating diseases, such as multiple sclerosis, but also in other highly prevalent diseases, such as Alzheimer's and psychosis. Current research into biology, structure and function of myelin is frequently hampered by the difficulty to assess this lipid-rich membrane structure in whole CNS (and PNS) preparations. Therefore, the technique presented here is highly welcome. I will focus here on the myelin and biology part of the manuscript as I am not a physicist.

The method presented here is an outstanding addition to existing methods such as MRI because it allows the assessment of the whole nervous system without dissection and other manipulation, in contrast to standard histological and immunohistological (including clearing) techniques. Especially in the diseased brain, this method will provide important advantages since in contrast to MRI, it is not susceptible to changes in tissue water content. Due to its exceptional sensitivity and specificity, it has the potential to be highly valuable for analyzing developmental myelination, dysmyelination as well as de- and remyelination in both experimental and human tissue.

The authors demonstrate an excellent validation of their technique in tissue samples. If available, imaging data from human cortex could further underline the exquisite sensitivity of the technique also in human tissue – with mostly more challenging tissue preparation than for experimental animals. Also, experimental models harboring more important tissue disruption compared to the shiverer mouse (e.g. with infiltration of inflammatory cells and different levels of axonal damage, etc.) could serve as proof that the method is well suited for detecting also pathologically distorted or newly formed, remyelinated myelin. However, I acknowledge that such experiments could also be included in later manuscripts.

Reviewer #2 (Remarks to the Author):

In the manuscript, "Nanostructure-specific X-ray tomography reveals myelin levels, integrity and axon orientations," Georgiadis et al. build on previous work using small-angle X-ray scattering (SAXS) to reconstruct the three-dimensional reciprocal space map of anisotropic, scattering biological samples. SAXS circumvents the scattering problem of deep biological imaging as X-rays have high penetrance even in very dense materials. The x-ray pencil beam is diffracted by periodic structures in the tissue and a 3D image can be reconstructed based on the 2D intensity diffraction maxima. Here, the authors exploit the periodicity of myelin wraps to generate macroscopic reconstructions of myelinated axons in nervous system samples from mice and humans. The incorporation of tensor tomography (TT) is an important advance with respect to past attempts to use SAXS for myelin imaging, as the method accounts for the anisotropy of the myelin structure and is the first that does

not rely on the assumption that SAXS signal intensity is independent of the sample orientation. SAXS-TT allows for the accurate quantification of myelin-specific signal and superior estimation of directional orientation of myelinated fibers. The results show proof of principle for this method to be applied in multiple nervous system samples (mouse brain, mouse spinal cord, human white matter), benchmark the reconstructions against both gold-standard MRI methods and contrasts (MT, dMRI, DTI/DKI, FA etc.) and histological serial reconstructions, and show a quantification of myelin content in a dysmyelinated vs. control mouse. Intriguingly, the authors can delineate PNS vs. CNS myelin based on the separation distance (d) of myelin wraps. This has important future implications for the investigation of compact vs. noncompact myelin, for example in de- vs. remyelinating lesions in multiple sclerosis.

Specific Issues:

1. Application in intact samples: The ability to apply this method to intact, living tissue would be a huge advance to the field. The data showing myelinated structures imaged through the intact skull presents an important foundation for this eventual advance. However, the authors state that, in their fixed tissue samples, "... comparison of MRI scans performed before and after the SAXS-TT experiments showed no effect of radiation, while the sample could further be processed... This suggests that radiation did not affect the tissue, at least regarding the length scales and the features probed by the mentioned methods." Since it is known that synchrotron radiation X-ray requires a high dose rate compared to normal X-Ray, and multiple studies have shown significant damaging effects on biological tissues and molecules (Chen et al., 2011), damaging effects on the tissue should not be discounted. Furthermore, the data from the before and after MRI scans mentioned above should be included in the manuscript rather than just mentioned.

2. Resolution and scan time: The reported resolution used in this study is relatively low compared to past applications of the technique. The first published x-ray microtomographic images used a pixel size of ~ 50 μm , yet here the reported resolution is $\sim 150\mu\text{m}$ (Table 1). With this resolution, the scan time for the mouse brain used was 87 hrs. A more in-depth discussion of the resolution and imaging parameters, as well as sample and system stability over multi-day scan times, would be beneficial to non-experts, especially for publication in a cross-disciplinary journal like Nature Communications. The low resolution is very evident in Figure 4g, although the imaging does seem to outperform the ex vivo diffusion MRI (dMRI) image shown in 4h. Still the representative images here bring up reservations about the method's ability to determine axonal and myelin microstructure.

3. Sample size: While the extremely difficult and time consuming methods presented in the current manuscript is not discounted, in all cases only a single mouse of each genotype and a single human white matter sample were used for SAXS-TT analyses. Although very small numbers of experimental subjects are common in whole-brain clearing studies (i.e. 2-4 brain samples, Chang et al., 2014), adding additional samples to these analyses would add significant information on the reproducibility and variability in the presented technique. Still, the presented data shows the first application of SAXS-TT to whole nervous system samples and represents a significant advance in noninvasive volumetric imaging.

4. Between samples comparisons: The authors state that, "It should be noted that orientations from tissue clearing and immunostaining were challenging to quantitatively compare across samples because of nonlinear tissue distortions, anisotropic voxels, inhomogeneous antibody penetration, high axonal density regions where signal appears homogeneous at optical microscopy resolution, etc." While decreased across-samples variability in whole brain white matter imaging would be a

major advance, the authors do not quantify this measure between the SAXS-TT samples. This Discussion point and the presentation of the method would be greatly improved by benchmarking the variability in SAXS-TT across samples. However, this quantification is not permitted by the data presented, since only single samples were used in the current analyses, see 3.

5. Myelin imaging specificity: The comparisons of SAXS-TT to Luxol-fast-blue stainings in Figure 4 and dysmyelinated Shiverer tissue in Figure 6 argue for sensitivity for detecting myelin content but does not evaluate the accuracy of myelinated axon orientations as stated in the manuscript title.

Comparisons of SAXS-TT to dMRI signal and the CLARITY-cleared neurofilament-stained mouse brain evaluate orientation of all axons but not specifically myelinated axons. Comparisons between SAXS-TT and CLARITY-cleared myelin basic protein-stained mouse brain (Chang et al., 2014) would allow the authors to make statements on myelinated axon orientation.

Reviewer #3 (Remarks to the Author):

In their manuscript, the authors report on the visualization of myelin using SAXS tensor tomography. Showing several applications like axon orientations in human and mouse brains (including a comparison to well-known methods including histology and MRI), investigations on CNS and PNS myelin or alterations in myelin levels, they underline the power of the technique. The article is well written and gives a deep insight. Due to the quality of the presented results, I recommend to publish the work in "Nature Communications". There are some minor points that need to be addressed before the appearance of the manuscript.

- Fig. 1: Looking on the scattering intensity image (d. i), one cannot distinguish between region 2 (gray matter) and region 5 (white matter). All other regions containing white matter have a much higher scattering intensity. How one can explain that? Is it due to the thickness of the brain at this position? Or small amount there?
- L. 109 Fig. 2e-f -> Fig. 2e,f
- Fig. 2a,b: For me it was confusing that the first two lines had different colors for the intensity as they show the same information (lower line after extracting the background. As red appears in the 3D renderings I suggest to use the 'hot' colormap for all the images.
- Fig. 2d: The box showing the position of the zoom in is hardly visible. Please either increase the line thickness or remove the box, as the location should be clear.
- Fig 2e: Zooming into the image showed that the quality of the image within the pdf is low (pixelated). I hope, the final version within the manuscript will be higher resolved.
- Fig. 3: If possible, I would split this figure into 2 figures and reorient the images line wise instead of column wise. All the images will get bigger and show the information without zooming into them.
- L. 154: Is the 10 um thin histological slice from the same brain, and is it even the corresponding SAXS-TT slice? It was not clearly stated in the text.
- Fig. 5b: First I would rearrange the images, first the scattering pattern, then the q-plots. For me it is not clear why two q-plots are shown in the figure. Both have the same information except the position of the yellow color below the lower curve. These two plots can easily be combined by using two different colors for CNS and PNS myelin
- L. 252: 'It can thereby avoid the tedious and artifacts-inducing sample preparation, sectioning and staining steps of histologic processing, while leaving the sample intact for further investigations.' It is obviously that SAXS-TT get additional information compared to conventional histology.

Nevertheless, the advantage of histology is that it can easily be performed in lab environment and faster than using SAXS-TT (taking into account acquisition time with the presented motor step size and processing time). I think it will be impossible to get beamtime for an extended study with a higher number of samples in order to get better statistics.

- L. 300: How were the MRI experiments performed, was the brain taken out the PBS-solution, was it in another liquid?

REVIEWER COMMENTS

Reviewer #1 (Remarks to the Author):

In the present work, the authors demonstrate a novel x-ray-based technique, SAXS-TT, for specific labeling of central and peripheral myelin sheaths in fixed tissue specimens *ex vivo* that utilizes a synchrotron-derived beam line. They present data on whole mouse brain, mouse spinal cord as well as human brain tissue, and validate their findings using different myelin-susceptible MRI weightings, standard Luxol-fast Blue histochemistry as well as tissue clearing and antibody stainings. Furthermore, they validate their level of myelin detection by using the dysmyelinating shiver mouse. They also exploit differences in CNS and PNS myelin nanostructure and periodicity to differentially display both types of myelin. They also show that their method is able to detect cortical myelin that is approximately 10 fold less dense than white matter myelin, demonstrating its usefulness in the settings of de- and dysmyelination.

Myelin structure and function are increasingly recognized to be affected not only in classical demyelinating diseases, such as multiple sclerosis, but also in other highly prevalent diseases, such as Alzheimer's and psychosis. Current research into biology, structure and function of myelin is frequently hampered by the difficulty to assess this lipid-rich membrane structure in whole CNS (and PNS) preparations. Therefore, the technique presented here is highly welcome. I will focus here on the myelin and biology part of the manuscript as I am not a physicist.

The method presented here is an outstanding addition to existing methods such as MRI because it allows the assessment of the whole nervous system without dissection and other manipulation, in contrast to standard histological and immunohistological (including clearing) techniques. Especially in the diseased brain, this method will provide important advantages since in contrast to MRI, it is not susceptible to changes in tissue water content. Due to its exceptional sensitivity and specificity, it has the potential to be highly valuable for analyzing developmental myelination, dysmyelination as well as de- and remyelination in both experimental and human tissue.

The authors demonstrate an excellent validation of their technique in tissue samples. If available, imaging data from human cortex could further underline the exquisite sensitivity of the technique also in human tissue – with mostly more challenging tissue preparation than for experimental animals. Also, experimental models harboring more important tissue disruption compared to the shiverer mouse (e.g. with infiltration of inflammatory cells and different levels of axonal damage, etc.) could serve as proof that the method is well suited for detecting also pathologically distorted or newly formed, remyelinated myelin. However, I acknowledge that such experiments could also be included in later manuscripts.

We thank the reviewer for the succinct review and kind words. Considering the constructive suggestions, we have performed additional experiments and added more samples to the study.

Following the reviewer's suggestion, we have scanned a human cortex specimen, in order to show the sensitivity and specificity of the method on such a sample too. The sample volume was $5 \times 14 \times 16 \text{ mm}^3$, the results are depicted in Supplementary Figure 3. We have specifically selected the V1 area (primary visual cortex), because this includes the "line of Gennari", a band

of myelinated axons within the cortex, whose myelin content has been a matter of investigation with tomographic methods. For instance, combined MRI and histology studies have shown that the low signal intensity of line of Gennari, and hence the contrast with regard to neighboring structures, is greatly influenced by its iron content (Duyn et al. 2007), and if the iron is removed the MRI contrast almost vanishes (Fukunaga et al. 2010) indicating a low contribution of myelin to the contrast. The myelin specificity of the SAXS-TT method allows investigating the integrity of the myelination of the fine structure. We have added a corresponding paragraph in the main text, and the appropriate mentions throughout the document (including the Methods section). In particular we have made the following additions to the manuscript.

1) Added Supplementary Figure 3:

Suppl. Fig. 3. SAXS-TT on human primary visual cortex (V1) of a 78yo female. a) The scanned specimen, with the line of Gennari pinpointed by the arrows. b) Characteristic diffraction pattern of the specimen, scanned in the 4-2 beamline of SLAC National Accelerator Laboratory. c) Virtual cross-section from the tensor tomographic reconstruction of the SAXS signal at the q -values of the myelin peak. d) Same virtual section depicting myelin levels from myelin-specific signal tensor tomographic reconstruction. e) Line plots across the green and magenta lines of (d), with line of Gennari indicated by arrows.

- 2) *Added the following text in the “Results” section, under the new sub-section title “SAXS-TT on human white matter and cortex”:*

*“We also applied SAXS-TT on a human cortex specimen of a 78yo female, **Suppl. Fig. 3**. We chose to image the V1 area (primary visual cortex) because this includes the line of Gennari, a band of myelinated fibers within the cortex that divides the deep and the superficial cortical layers. The contrast between the line of Gennari and the surrounding cortex in tomographic methods such as MRI (Kleinnijenhuis et al. 2013) has been also attributed to iron (Duyn et al. 2007), since iron removal minimizes the contrast (Fukunaga et al. 2010). In the selected specimen, the myelinated line was in regions visible to the bare eye (Suppl. Fig. 3a), and could be clearly distinguished in the tomographic SAXS signal reconstruction at the q -values of the myelin peak (Suppl. Fig. 3c). The band was less clearly visible when visualizing the myelin levels based on the myelin-specific signal reconstruction (Suppl. Fig. 3d), due to the significantly lower myelin level values of the line compared to the subcortical white matter. This finding is expected given the advanced age of the donor, which contributes to decreased myelin levels at the line of Gennari (Lintl and Braak 1983). Moreover, the outer cortical layers contained almost no myelin and are barely visible in the myelin map, whereas the deeper cortical layers were more myelinated. These observations are also corroborated by the quantitative values shown in the line plots of Suppl. Fig. 3e: myelin levels decrease sharply going from white matter to the cortex, increase minimally at the position of the line of Gennari, and then decrease to almost zero at the outer cortical layers, in line with the trend that can be observed qualitatively by myelin histology (Lintl and Braak 1983; Balaram, Young, and Kaas 2014; Fukunaga et al. 2010).*

Overall, the myelin specificity of the SAXS-TT method allows quantitatively investigating the myelination within and across human brain specimens, including fine myelinated structures.”

- 3) *We added the following text in the “Methods” section:*

“The human primary visual cortex sample was excised from a formalin-fixed brain of a 78 year-old female with no pathological finding in the cortex, from the tissue bank of Stanford’s Alzheimer’s Disease Research Center (ADRC). The brain was preserved in 1% PBS + 0.02% sodium azide solution for approximately 3 years.”

- 4) *Moreover, since the experiment was performed in a different synchrotron than the previous experiments (in Stanford Synchrotron Radiation Lightsource, in the SLAC National Accelerator Laboratory), Supplementary Figure 1 has changed accordingly, to include a sub-figure of the SAXS-TT setup in the SSRL 4-2 beamline:*

Suppl. Fig. 1. SAXS-TT setup in 3 synchrotrons.

a) SAXS-TT setup in cSAXS beamline, at the Swiss Light Source synchrotron, in the Paul Scherrer Institute, Villigen, Switzerland. **b)** SAXS-TT setup in LiX beamline, (DiFabio et al. 2016) at National Light Synchrotron Source II, in the Brookhaven National Laboratory, New York, USA. **c)** SAXS-TT setup in 4-2 beamline, at the Stanford Synchrotron Radiation Lightsource synchrotron, in the SLAC National Accelerator Laboratory, California, USA. An undulator (cSAXS, LiX) or a wiggler (4-2) produce a very intense X-ray beam, which is focused and shaped by X-ray optics, and sent to interact with the sample (dashed red line). Motorized translation and rotation stages realize the raster scanning of the sample sitting in a Kapton tube, at multiple projections around one (b, c) or two (a) rotation axes. The photons that have interacted with the sample and are scattered at small angles travel through an evacuated “flight tube” and are collected by a photon-counting Pilatus detector (Henrich et al. 2009) ~2-3m downstream. The direct beam is collected by a photodiode, which both blocks it from reaching the photon-sensitive detector and measures it to obtain transmission information.

We fully agree with the remark of the reviewer that further quantitative group studies on pathologic nervous tissue are indeed of very high interest: we are currently performing such studies and plan to include them in future manuscripts, as the reviewer acknowledges.

Reviewer #2 (Remarks to the Author):

In the manuscript, "Nanostructure-specific X-ray tomography reveals myelin levels, integrity and axon orientations," Georgiadis et al. build on previous work using small-angle X-ray scattering (SAXS) to reconstruct the three-dimensional reciprocal space map of anisotropic, scattering biological samples. SAXS circumvents the scattering problem of deep biological imaging as X-rays have high penetrance even in very dense materials. The x-ray pencil beam is diffracted by periodic structures in the tissue and a 3D image can be reconstructed based on the 2D intensity diffraction maxima. Here, the authors exploit the periodicity of myelin wraps to generate macroscopic reconstructions of myelinated axons in nervous system samples from mice and humans. The incorporation of tensor tomography (TT) is an important advance with respect to past attempts to use SAXS for myelin imaging, as the method accounts for the anisotropy of the myelin structure and is the first that does not rely on the assumption that SAXS signal intensity is independent of the sample orientation. SAXS-TT allows for the accurate quantification of myelin-specific signal and superior estimation of directional orientation of myelinated fibers. The results show proof of principle for this method to be applied in multiple nervous system samples (mouse brain, mouse spinal cord, human white matter), benchmark the reconstructions against both gold-standard MRI methods and contrasts (MT, dMRI, DTI/DKI, FA etc.) and histological serial reconstructions, and show a quantification of myelin content in a dysmyelinated vs. control mouse. Intriguingly, the authors can delineate PNS vs. CNS myelin based on the separation distance (d) of myelin wraps. This has important future implications for the investigation of compact vs. noncompact myelin, for example in de- vs. remyelinating lesions in multiple sclerosis.

We would like to thank the reviewer for the detailed comments and suggestions. We have addressed the suggested changes as described point-by-point below.

Specific Issues:

1. Application in intact samples: The ability to apply this method to intact, living tissue would be a huge advance to the field. The data showing myelinated structures imaged through the intact skull presents an important foundation for this eventual advance. However, the authors state that, in their fixed tissue samples, "... comparison of MRI scans performed before and after the SAXS-TT experiments showed no effect of radiation, while the sample could further be processed... This suggests that radiation did not affect the tissue, at least regarding the length scales and the features probed by the mentioned methods." Since it is known that synchrotron radiation X-ray requires a high dose rate compared to normal X-Ray, and multiple studies have shown significant damaging effects on biological tissues and molecules (Chen et al., 2011), damaging effects on the tissue should not be discounted. Furthermore, the data from the before and after MRI scans mentioned above should be included in the manuscript rather than just mentioned.

The reviewer is very correct in pointing out that the dose imparted on the tissue might have damaging consequences which should not be discounted. To that end, we have rephrased the corresponding discussion text, and also included figures from the pre- and post-MRI scans. We have additionally included data from the very first SAXS-TT projection and a repetition of the

same projection at the very end of the scan, which we systematically acquire in our experiments precisely in order to be able to identify any effects from the scan.

We have accordingly revised the respective comment in the "Discussion" section:

Previous version:

"Limitations of the method include the current incompatibility of SAXS-TT experiments with *in vivo* investigations due to radiation considerations, although synchrotron X-ray methods become increasingly compatible with *in vivo* studies.³¹ In addition, comparison of MRI scans performed before and after the SAXS-TT experiments showed no effect of radiation, while the sample could further be processed for 2D or 3D histology, stained and imaged, as presented above. This suggests that radiation did not affect the tissue, at least regarding the length scales and the features probed by the mentioned methods."

New version:

"Limitations of the method include the radiation dose imparted by X-rays, which may result in significant damage on biological tissue (Asaithamby, Hu, and Chen 2011). However, the imparted dose and dose rate in the presented experiments seemed to be in a range that did not affect the tissue morphology at the investigated length scales: first, scan of the same sample projection at the beginning and the end of the X-ray experiment (**Suppl. Figure 7a**) showed identical X-ray scattering signals. Second, MRI scans performed with the same parameters before and after SAXS-TT (Suppl. Figure 7b) did not reveal detectable contrast differences -minor anatomical discrepancies are due to minimal changes in slice position. It should be additionally noted that the scanning is fully compatible with further processing of the sample, for instance for 2D and 3D histology, staining and imaging as presented above. Moreover, synchrotron X-ray methods become increasingly compatible with *in vivo* investigations (Morgan et al. 2020), a direction we aim to pursue in future studies."

We have also added a paragraph in the "Methods" section to describe the MRI sequences resulting in the figures depicted in Suppl. Figure 7b:

"For the anatomical maps in Suppl. Fig. 7b, used for pre- and post-scan comparison, the following sequences were used: i) top row: T_2 -weighted Bruker RARE sequence, 1st TE: 6.1ms, effective TE: 48.7ms, rare factor: 16, TR=2s, isotropic voxel size 100 μ m, matrix 150 \times 200 \times 80 (=15 \times 20 \times 8mm³), scan time: 16m. ii) bottom row: T_1 -weighted Bruker FLASH (Fast Low Angle Shot) sequence, TE=4.5ms, TR=250ms, flip angle: 15 $^\circ$, isotropic voxel size 100 μ m, matrix 170 \times 110 \times 67 (=17 \times 11 \times 6.7mm³), scan time: 33m27s."

Supplementary Figure 7:

Suppl. Fig. 7. X-ray and MRI data before and after the SAXS-TT scan of the C57BL/6 mouse brain sample. **a)** SAXS projections for sample orientation $(\beta, \alpha) = (0^\circ, 0^\circ)$ at the beginning (first projection -left) and the end (after the last projection -right) of the SAXS-TT scan. Inset: Scattering intensity- q plot for the same sample point in the beginning and the end of the scan (marked by a star in the projection figures). **b)** T_2 - and T_1 -weighted MRI scans (top and bottom row respectively) acquired a few days before (left) and after (right) the SAXS-TT scan.

2. Resolution and scan time: The reported resolution used in this study is relatively low compared to past applications of the technique. The first published x-ray microtomographic images used a pixel size of $\sim 50\ \mu\text{m}$, yet here the reported resolution is $\sim 150\ \mu\text{m}$ (Table 1). With this resolution, the scan time for the mouse brain used was 87 hrs. A more in-depth discussion of the resolution and imaging parameters, as well as sample and system stability over multi-day scan times, would be beneficial to non-experts, especially for publication in a cross-disciplinary journal like Nature Communications. The low resolution is very evident in Figure 4g, although the imaging does seem to outperform the ex vivo diffusion MRI (dMRI) image shown in 4h. Still the representative images here bring up reservations about the method's ability to determine axonal and myelin microstructure.

The reviewer correctly points out that the resolution presented here is lower than in previous experiments. However, this is due to the significant increase of the sample size.

For instance, in Liebi et al., Nature, 2015, sample size was $\sim 1 \times 1 \times 2.5\ \text{mm}^3$, with a resolution of $25\ \mu\text{m}$, and in Schaff et al., Nature, 2015, sample size was $\sim 3 \times 3 \times 4\ \text{mm}^3$, with a resolution of $50\ \mu\text{m}$. In the presented experiments, brain specimens are ~ 10 - and ~ 3 -fold larger respectively, and the resolution is correspondingly lower, so that the scan times are in the same range, of some tens of hours per specimen.

We have added a discussion on the imaging parameters and scan stability over long experimental sessions, as suggested by the reviewer. We share the view that this is beneficial for non-experts, and we believe it helps improve the manuscript.

We made the following change to the manuscript ("Discussion" section):

*"A challenge of SAXS-TT experiments is the long scan times needed (cf. **Table 1** in Methods). Scan duration depends on the target resolution, with the scan duration being -approximately- inversely proportional to the cube of the voxel dimension. For instance, decreasing the (isotropic) voxel size from $200\ \mu\text{m}$ to $100\ \mu\text{m}$ -while keeping the exposure time the same- will induce an ~ 8 -fold increase in scan time. The multi-hour scans needed to cover larger samples with adequate resolution might strain the sample and system stability. To that effect, the beam intensity should be tracked and taken into account, as explained in the Methods section. Also, we have observed that higher agarose gel concentrations ($> 1\% \text{ w/v}$) result in gel evaporation and the formation of bubbles during the scans, and should thus be avoided."*

3. Sample size: While the extremely difficult and time consuming methods presented in the current manuscript is not discounted, in all cases only a single mouse of each genotype and a single human white matter sample were used for SAXS-TT analyses. Although very small numbers of experimental subjects are common in whole-brain clearing studies (i.e. 2-4 brain samples, Chang et al., 2014), adding additional samples to these analyses would add significant information on the reproducibility and variability in the presented technique. Still, the presented data shows the first application of SAXS-TT to whole nervous system samples and represents a significant advance in noninvasive volumetric imaging.

4. Between samples comparisons: The authors state that, "It should be noted that orientations from tissue clearing and immunostaining were challenging to quantitatively compare

across samples because of nonlinear tissue distortions, anisotropic voxels, inhomogeneous antibody penetration, high axonal density regions where signal appears homogeneous at optical microscopy resolution, etc.” While decreased across-samples variability in whole brain white matter imaging would be a major advance, the authors do not quantify this measure between the SAXS-TT samples. This Discussion point and the presentation of the method would be greatly improved by benchmarking the variability in SAXS-TT across samples. However, this quantification is not permitted by the data presented, since only single samples were used in the current analyses, see 3.

We acknowledge that the current manuscript aims to presents a new method and its first application on different human and mouse nervous system samples. We are aware that an endeavor to benchmark the across-sample variability would involve multiple samples (eg. mouse brains of the same strain) examined with SAXS-TT as well as with a reference method (eg. histology), so that the across-sample variability could be quantified and compared to the current “gold standard”. This is a great study suggestion, which we aim to perform and include in future manuscripts. In fact, we are already performing experiments on groups of control and pathologic brain samples, where across-sample variability will be quantified, and will include these in future manuscripts.

In order to address the reviewer’s suggestions and quantify the method variability within this study, we have added more samples, as well as performed additional analyses:

- 1. Since the mammalian brain provides us with a system inherently containing two almost identical sub-systems (the two hemispheres), we quantified and compared myelin levels in the two hemispheres of the C57BL/6 mouse brain, for multiple regions of interest. We have visualized this analysis in a bar graph, which is included in Figure 2 of the manuscript (Figure 2h). The analysis revealed that myelin values for identical structures of the two hemispheres are almost identical, with myelin levels in all white matter regions being statistically similar, as confirmed by an unpaired t-test between values of the left and right hemisphere in each ROI:*

Figure 1. SAXS-TT results for mouse brain. **a)** Sagittal, axial and coronal virtual slices of tensor-reconstructed reciprocal space-map intensity at myelin peak q -values. Grayscale colormap was used for the unspecific scattering signal, akin to unspecific tomographic contrasts such as MRI or CT. **b)** Same virtual slices depicting the tensor-reconstructed myelin-specific signal. “Hot” colormap was used for the myelin-specific signal, akin to molecular imaging outcomes, e.g. using fluorescent tags. **c)** 3D myelin distribution map of highly myelinated areas (red). **d)** Coronal slice of the 3D fODF map, with tensors represented by ellipsoids, colored by tensor trace. **e)** Side-view of 3D map of fiber orientations, represented by lines. Color and orientation correspond to the largest tensor eigenvalue. **f)** Same coronal slice as in (a,b,d) from map in (e). **g)** Tractogram of same section, based on main fiber orientation. **h)** Bar graph displaying mean myelin levels and standard deviations for different regions of the mouse brain, for right and left hemisphere. All white matter regions (CIN, AC, CC, CP, CST,

IC, CBP, OPT) displayed statistically similar myelin levels between left and right hemispheres using an unpaired t-test. [VEN: lateral ventricles, p-value: 10^{-5} | CRUS: crus 1&2, cerebellar lobule 7 (gray matter), p-value: $2 \cdot 10^{-4}$ | OBGL: olfactory bulb, granule layer (gray matter), p-value: $2 \cdot 10^{-3}$ | CIC: cingulate cortex, p-value:0.016 | AMY: amygdala, p-value:0 | SSC: secondary somatosensory cortex, p-value: $3 \cdot 10^{-9}$ | PVC: primary visual cortex, p-value:0.048 | BF: barrel field, p-value: $2 \cdot 10^{-14}$ | HP: hippocampus, p-value:0 | HTH: hypothalamus, p-value: $4 \cdot 10^{-16}$ | STR: striatum, p-value: $2 \cdot 10^{-13}$ | TH: thalamus, p-value:0 | GLP: globus pallidus, p-value:0.28 | CIN: cingulum, p-value:0.58 | AC: anterior commissure, p-value:0.43 | CC: corpus callosum, p-value:0.81 | CP: cerebral peduncle, p-value:0.17 | CST: cerebrospinal tract, p-value:0.54 | IC: internal capsule, p-value:0.84 | CBP: cerebellar peduncle, p-value:0.21 | OPT: optic tract, p-value:0.70]

2. Moreover, we have included a second human white matter corpus callosum sample from the same subject, and quantified variability within and across the two samples. This is now reflected in the new Figure 4:

Figure 4. SAXS-TT on corpus callosum (CC) splenium and body specimens of 2-year-old female. (1) Samples, in red circle, were raster scanned and **(2)** produced strong myelin peaks in their diffraction patterns (white arrows). **(3)** Each 2D scan resulted in a projection depicting SAXS myelin peak intensity and 2D orientation of the myelinated axons. **(4)** Tensor-tomographic reconstruction of all projections provided 3D fiber orientations per voxel, here visualized by vector maps for virtual sections. Vectors are colored based on myelin levels in the corresponding voxel. **(5)** Tractography algorithms enabled generating neuronal tracts for the two specimens, representative sections of which are shown here. The bar graph at the bottom of the figure shows the myelin levels -and respective standard deviations- of the specimens, with the body displaying significantly higher values. Significance p-value of the unpaired t-test and the Kolmogorov-Smirnov test is below 10^{-10} .

We have added the following text commenting on the myelin level comparison of the two regions:

"The bar graph at the bottom of Fig. 4 displays the myelin levels for both samples. The levels at the body were significantly higher than at the splenium of the corpus callosum, consistent with myelination patterns observed in developing brains assessed with postmortem pathology (Brody et al. 1987; Kinney et al. 1988)."

3. Finally, we have added the following discussion on the topic:

"When it comes to comparisons within and across samples, specific care has to be taken when performing quantitative analyses. Within a single sample or across samples scanned in the same beamtime, the fluctuations of the incoming X-ray beam during the experiment have to be taken into account. This can be achieved either by a dedicated sensor constantly measuring the incoming beam, or by the off-sample intensity measured by the beamstop in every line of scanning. For comparisons between samples across different beamtimes or beamlines, quantification of the SAXS intensity is needed, which is possible by using SAXS calibration standards (Allen et al. 2017). This approach will also enable benchmarking the intra- and inter-sample variability of the method across multiple samples, beamtimes and beamlines, extending the variability analyses presented in Figs. 2 and 4."

5. Myelin imaging specificity: The comparisons of SAXS-TT to Luxol-fast-blue stainings in Figure 4 and dysmyelinated Shiverer tissue in Figure 6 argue for sensitivity for detecting myelin content but does not evaluate the accuracy of myelinated axon orientations as stated in the manuscript title. Comparisons of SAXS-TT to dMRI signal and the CLARITY-cleared neurofilament-stained mouse brain evaluate orientation of all axons but not specifically myelinated axons. Comparisons between SAXS-TT and CLARITY-cleared myelin basic protein-stained mouse brain (Chang et al., 2014) would allow the authors to make statements on myelinated axon orientation.

The reviewer accurately highlights that the dMRI and CLARITY datasets do not specifically examine myelinated axon orientations, but overall microstructural and axonal orientations respectively. While this is stated for the dMRI dataset in the discussion ("... and to SAXS-TT and dMRI measuring very different biophysical phenomena and micro-/nano-structural characteristics: SAXS-TT is sensitive to myelinated axons only, while dMRI signals comprise contributions from multiple structures and intra- or extra-axonal compartments.²⁹"), it is currently not

clear for the CLARITY dataset. We have changed the discussion text accordingly to clarify this distinction:

"Moreover, neurofilament staining did not stain myelinated axons only, but all axons. A myelinated-axon-specific stain, e.g. targeting myelin basic protein, would allow such quantification, though it could not be used on models such as shiverer mice that lack the protein. All these further highlight the need for a reference method for myelinated axon orientations in macroscopic samples, a role we suggest could be assumed by the presented SAXS-TT-based method."

Reviewer #3 (Remarks to the Author):

In their manuscript, the authors report on the visualization of myelin using SAXS tensor tomography. Showing several applications like axon orientations in human and mouse brains (including a comparison to well-known methods including histology and MRI), investigations on CNS and PNS myelin or alterations in myelin levels, they underline the power of the technique. The article is well written and gives a deep insight. Due to the quality of the presented results, I recommend to publish the work in "Nature Communications". There are some minor points that need to be addressed before the appearance of the manuscript.

We thank the reviewer for the positive stance towards our study and the detailed suggestions to improve the manuscript. We have tried to address them point-by-point as below:

- Fig. 1: Looking on the scattering intensity image (d. i), one cannot distinguish between region 2 (gray matter) and region 5 (white matter). All other regions containing white matter have a much higher scattering intensity. How one can explain that? Is it due to the thickness of the brain at this position? Or small amount there?

As the reviewer correctly guessed, this is due to the thickness of the brain tissue at this point. As Figures 1b,c show, myelin contributes a peak on top of a non-specific signal from the tissue. Even areas with no or very little myelin (sub-figure c2) have much more signal at the investigated q-range than the background medium (sub-figure c1). As a result, the total signal from these areas along the beam path can be similar to the total signal from highly myelinated areas where there is a lot less tissue along the beampath.

To make that more clear for the readers, we have made the following addition to the figure caption:

Previous version:

"... d) 2D maps for one projection."

New version:

"... d) 2D maps for one projection, based on the signal along the beampath for each point."

- L. 109 Fig. 2e-f -> Fig. 2e,f

The change was made following the reviewer's suggestion.

- Fig. 2a,b: For me it was confusing that the first two lines had different colors for the intensity as they show the same information (lower line after extracting the background. As red appears in the 3D renderings I suggest to use the 'hot' colormap for all the images.

The reviewer is reasonably confused by the choice of different colors. The specific colormaps were used on purpose, with the following reasoning:

The top row is the reconstruction of the overall, unspecific signal, so the colormap used is the same as in typical images of non-specific tomographic modalities, such as CT or MRI.

The bottom row is the myelin-specific signal, so the color map used resembles the one typically used in molecular imaging, e.g. when using antibody labeling followed by fluorescence imaging.

The red color in the 3D rendering is also chosen because it represents myelin specifically.

The only case where we have not used the "hot" colormap to show cross-sections of myelin maps is in Figure 5, where we retrieve both central and peripheral myelin. In that case, since the maps show 2 different myelin types, we have used magenta and green, two colors that would enable color-blind people to distinguish the two.

This clarification on the color used was added to the caption of Figure 2:

Previous version:

"Figure 2. SAXS-TT results for mouse brain. a) Sagittal, axial and coronal virtual slices of tensor-reconstructed reciprocal space-map intensity at myelin peak q-values. b) Same virtual slices depicting the tensor-reconstructed myelin-specific signal."

New version:

"Figure 2. SAXS-TT results for mouse brain. a) Sagittal, axial and coronal virtual slices of tensor-reconstructed reciprocal space-map intensity at myelin peak q-values. Grayscale color map was used for the unspecific scattering signal, akin to unspecific tomographic contrasts such as MRI or CT. b) Same virtual slices depicting the tensor-reconstructed myelin-specific signal. "Hot" colormap was used for the myelin-specific signal, akin to molecular imaging outcomes, e.g. using fluorescent tags."

- Fig. 2d: The box showing the position of the zoom in is hardly visible. Please either increase the line thickness or remove the box, as the location should be clear.

The box outline has been made thicker to improve visibility, we thank the reviewer for the suggestion.

- Fig 2e: Zooming into the image showed that the quality of the image within the pdf is low (pixelated). I hope, the final version within the manuscript will be higher resolved.

The low image quality is indeed due to the pdf conversion in the submission process, the final version should be of high resolution.

- Fig. 3: If possible, I would split this figure into 2 figures and reorient the images line wise instead of column wise. All the images will get bigger and show the information without zooming into them.

We thank the reviewer for the suggestion, which will enable readers to better appreciate the manuscript figures. We have now indeed split the two samples in different figures. The human white matter splenium sample now appears in Figure 4, in a line-wise format as suggested by the reviewer; also taking into account comments of the other reviewers, we have added a second human white matter sample in the same figure, always in a line-wise format as suggested by the reviewer, and taking care so that the figures are of adequate size to be appreciated without zooming in, to the extent possible.

CC splenium

1. Sample

2. Diff. pattern

3. Projections

4. SAXS-TT reconstruction

5. Tracts

CC body

1. Sample

2. Diff. pattern

3. Projections

4. SAXS-TT reconstruction

5. Tracts

We are also including the spinal cord sample as a separate figure, Figure 3. In that case, we kept the same format as before but have condensed the figure in the horizontal direction so that the individual sub-figures are enlarged when fitted to the page, and thus more clearly visible:

- L. 154: Is the 10 μm thin histological slice from the same brain, and is it even the corresponding SAXS-TT slice? It was not clearly stated in the text.

The histological slices are from the left hemisphere of the same brain scanned with SAXS-TT. Although this is made clear in the Methods (“... After two more months in 1% PBS at 4oC, the brain was cut in half at the mid-sagittal plane. The left hemisphere was sent for histology sectioning and myelin staining. ...”), it is not clear in the main text, so we have added the word “same” in the mentioned sentence to indicate that.

Previous version:

“... we compared the SAXS-TT-derived myelin levels for the C57BL/6 mouse brain with bright-field images of 320 consecutive Luxol-fast-blue-stained 10 μm -thick histological sections covering the brain’s left hemisphere.”

New Version:

“... we compared the SAXS-TT-derived myelin levels for the C57BL/6 mouse brain with bright-field images of 320 consecutive Luxol-fast-blue-stained 10 μm -thick histological sections covering the same brain’s left hemisphere.”

Similarly, we have adapted the caption wording to reflect the fact that the shown sections are the same ones with different contrasts, registered to each other:

Previous version:

"Figures below table depict sagittal slice of parameter maps."

New Version:

"Figures below table depict the same sagittal slice of the compared parameter maps."

- Fig. 5b: First I would rearrange the images, first the scattering pattern, then the q-plots. For me it is not clear why two q-plots are shown in the figure. Both have the same information except the position of the yellow color below the lower curve. These two plots can easily be combined by using two different colors for CNS and PNS myelin

The reviewer is correct in pointing out that the CNS and PNS q-plots' difference is the highlighted part, and that logically the diffraction pattern should go to the left and the resulting q-plots to the right. The reason the figure had not been implemented as suggested was that we wanted to be consistent throughout the Figure: in all of a, b, c, d, the left-most (virtual) column had CNS information, the central one PNS information, and the rightmost was integrating both CNS and PNS.

However, as mentioned, we agree that the suggestion of the reviewer follows the logical flow, and for this reason we have implemented the changes as suggested, while also highlighting the color pattern throughout the figure (CNS →magenta, PNS →green):

- L. 252: 'It can thereby avoid the tedious and artifacts-inducing sample preparation, sectioning and staining steps of histologic processing, while leaving the sample intact for further investigations.' It is obviously that SAXS-TT get additional information compared to conventional histology. Nevertheless, the advantage of histology is that it can easily be performed in lab environment and faster than using SAXS-TT (taking into account acquisition time with the presented motor step size and processing time). I think it will be impossible to get beamtime for an extended study with a higher number of samples in order to get better statistics.

The reviewer is correct to highlight the wide availability of histology. With our study we aim to showcase the advantages of SAXS-TT (specific, quantitative, non-destructive scanning, for quantitative comparisons within and across samples). We understand that the method will not replace histology, but we hope that our manuscript will encourage researchers interested in myelin and myelinated axon orientation quantification, and will result in a wide use of the technology for studies that need to thoroughly examine these features.

The reviewer is also correct to point to the challenges of obtaining long beamtime needed for an extended study. While this is certainly true, beamlines are increasingly capable of performing high-throughput experiments, and we anticipate an increase in demand for such experiments that will further accelerate the process. It should be mentioned that we are currently in the process of performing such a study "with a higher number of samples in order to get better statistics" on healthy versus pathological human brain tissue, which we hope to present in the near future.

We have added the following text to discuss this very interesting point:

"Moreover, long SAXS-TT acquisition times require allocation of long synchrotron beam times for studies with multiple samples. Yet, beamlines are increasingly capable of accommodating multi-sample experiments, due to the implementation of on-the-fly scanning, combined with high-sensitivity detectors and higher fluxes available in modern synchrotrons."

- L. 300: How were the MRI experiments performed, was the brain taken out the PBS-solution, was it in another liquid?

This was in fact not clear in the manuscript - we thank the reviewer for pointing that out. To that end, we have added the following text in a prominent position in the Methods (just under the MRI experiments heading):

"All samples were immersed in a perfluorocarbon solution (Fomblin®) during MRI scanning."

Reviewer #2 (Remarks to the Author):

The authors have adequately addressed the suggested changes. On the topic of orientation of myelinated fibers, future work confirming the hypothesis that the SAXS-TT-based method can be used for definitive analyses of this parameter will be well received.

Reviewer #3 (Remarks to the Author):

The revision of the manuscript was prepared carefully. The authors responded to all my comments point-by-point reasonably. From my side, the manuscript can be published as it is.

Point-by-point response to Reviewers' comments

For manuscript "*Nanostructure-specific X-ray tomography reveals myelin levels, integrity and axon orientations in human and mouse nervous tissue*" by Georgiadis et al.

REVIEWERS' COMMENTS

Reviewer #2 (Remarks to the Author):

The authors have adequately addressed the suggested changes. On the topic of orientation of myelinated fibers, future work confirming the hypothesis that the SAXS-TT-based method can be used for definitive analyses of this parameter will be well received.

We thank the reviewer for his previous and current feedback. To address the suggestion for the future work, we have now added these phrases in the discussion:

"Myelin-specific stains, e.g. targeting myelin basic protein (MBP) or myelin proteolipid protein (PLP), would allow such quantification, and will be used in future studies aiming at direct quantitative comparisons"

Reviewer #3 (Remarks to the Author):

The revision of the manuscript was prepared carefully. The authors responded to all my comments point-by-point reasonably. From my side, the manuscript can be published as it is.

We thank the reviewer for all his comments and help in making the manuscript clearer.